# A comprehensive benchmarking with interpretation and operational guidance for the hierarchy of topologically associating domains

Jingxuan Xu[1,8], Xiang Xu[2,8], Dandan Huang[3,4,8], Yawen Luo[2], Lin Lin[2,5], Xuemei Bai[2], Yang Zheng[2], Qian Yang[2], Yu Cheng[1], An Huang[1], Jingyi Shi[1], Xiaochen Bo ®[2] ✉, Jin Gu ®[1,3,4,6,7] ✉ & Hebing Chen ®[2] ✉

Topologically associating domains (TADs), megabase-scale features of chromatin spatial architecture, are organized in a domain-within-domain TAD hierarchy. Within TADs, the inner and smaller subTADs not only manifest cell-to-cell variability, but also precisely regulate transcription and differentiation. Although over 20 TAD callers are able to detect TAD, their usability in biomedicine is confined by a disagreement of outputs and a limit in understanding TAD hierarchy. We compare 13 computational tools across various conditions and develop a metric to evaluate the similarity of TAD hierarchy. Although outputs of TAD hierarchy at each level vary among callers, data resolutions, sequencing depths, and matrices normalization, they are more consistent when they have a higher similarity of larger TADs. We present comprehensive benchmarking of TAD hierarchy callers and operational guidance to researchers of life science researchers. Moreover, by simulating the mixing of different types of cells, we confirm that TAD hierarchy is generated not simply from stacking Hi-C heatmaps of heterogeneous cells. Finally, we propose an air conditioner model to decipher the role of TAD hierarchy in transcription.

Dramatic advances in high-throughput sequencing of genome-wide interactions (Hi-C) have brought high-order genome spatial architecture within 2-μm nucleus to light, spanning from chromosome territories, A/B compartments to megabase-scale topologically associating domains (TADs) and chromatin loops[1–3]. Accompanied by numerous studies on embryogenesis, lineage differentiation, and various diseases[4,5], TADs are revealed to serve as principal units of chromatin folding and gene regulation[2,6], ranging from 100 kb to 1 Mb. Initially recognized as the fractal globule model and blocks along diagonal on Hi-C heatmap[1], TAD manifests high self-interaction within itself and insulates contacts with other TADs by convergent CCCTC-binding factor (CTCF) binding and cohesin combination[2,7–10], known as 'loop extrusion' model[11,12], confining appropriate interactions between regulatory elements and target genes[13]. Although TADs are

[1]Key Laboratory of Carcinogenesis and Translational Research (Ministry of Education/Beijing), Department of Gastrointestinal Surgery, Peking University Cancer Hospital & Institute, Beijing 100142, China. [2]Academy of Military Medical Science, Beijing 100850, China. [3]Department of Oncology, Peking University Shougang Hospital, Beijing, China. [4]Center for Precision Diagnosis and Treatment of Colorectal Cancer and Inflammatory Diseases, Peking University Health Science Center, Beijing, China. [5]School of Computer Science and Information Technology& KLAS, Northeast Normal University, Changchun, China. [6]Peking-Tsinghua Center for Life Sciences, Peking University, Beijing, China. [7]Peking University International Cancer Institute, Beijing, China. [8]These authors contributed equally: Jingxuan Xu, Xiang Xu, Dandan Huang. ✉e-mail: boxc@bmi.ac.cn; zlguj@bjmu.edu.cn; chb-1012@163.com

conserved across cell types and species[2,14–16], insertion or deletion of TAD boundaries would give rise to ectopic enhancer-promoter contacts, leading to abnormal gene expression, development disability, and tumorigenesis[4,17,18]. Similarly, TAD reorganization through cell fate transition is high correlated with gene expression changes[14,19–21].

It's previously found that TADs could be further subdivided into subTADs[7,22,23], and the Phillips group systematically detected subTADs with directional index adapted by a Hidden Markov Model (HMM-DI) in high-resolution 5 C maps[2,22]. Perhaps subTADs could account for the discrepancy in size and number of TADs from different TAD callers, multiple resolutions, and matrices normalization[12,24]. SubTADs show more dynamics than TADs, including cell-to-cell variation and tissue-to-tissue variation, and generate insulated neighborhoods within a TAD[20,25,26], retaining the independence of gene expression. Increasingly number of researches focus on these smaller subTADs in embryonic development, lineage differentiation, and tumorigenesis[27–32]. Even manipulating just these tens-of-kilobase scale subTADs enables precise control of transcription and phenotypes[33,34]. Apart from CTCF or cohesin, subTAD boundaries are discovered to be confined by active epigenomic markers (indicating strong enhancers and active promoters), mediators, transcription factors, transcription start sites (TSSs), and transcription termination sites (TTSs)[35], which define the organization, maintenance or reconfiguration of certain subTADs[29–32,34]. Such diverse anchors of subTAD boundaries may contribute to highly dynamic nature and weakness compared with TAD boundaries[25]. In addition to the formation mechanism mentioned above, subTADs take shape before convergence into hierarchical TAD structures during the mitosis-to-G1 phase transition, indicating a bottom-up formation of TAD/subTAD[36]. Coincidentally, chromatin has been revealed to be organized into a domains-within-domains tree-like hierarchy, nominated metaTAD, which extends the maximum size of TAD domains to 2 Mb And units in metaTAD tend to be reorganized through neuron differentiation, associated with transcription state transition[19,37]. Recent research collectively refers to TAD, subTAD, and metaTAD as TAD hierarchy, with the level characterized by positive correlations with CTCF enrichment, gene activity, gene density, and active epigenetic states. In contrast, single TAD conveys the lowest extent of these aspects[38]. Furthermore, the hierarchical level of TAD boundary remarkably separates colorectal carcinoma from normal colon, which is relevant to novel transcription and prognosis[39]. The 'asymmetric extrusion' model is proposed as a possible formation mechanism of TAD hierarchy, according to distinct epigenetic features of adjacent subTADs[38]. Together, TAD hierarchy is necessary to explore gene regulation and diseases, of which increasingly more functions and applications are hopefully to be discovered in follow-up research.

Obviously, TAD hierarchy expands the definition and size range of typical TADs. However, a few critical problems remain to be solved: whether TAD hierarchy is generated simply from stacking Hi-C heatmaps of heterogeneous cells, what the mechanism of its formation is, and how it acts on gene regulation. Besides, confusion arises when picking out tools for detecting TAD hierarchy. In this work, we compared the performance of 13 computational methods for TAD hierarchy prediction in robustness (matrices-normalization, various resolutions, and various sequencing depths), epigenomic features, and tools usability. Hierarchy similarity is evaluated by a metric we developed and one metric previously reported[40]. Among these methods, we hope to give biomedical researchers tools-recommendation for appropriate datasets or special studies and convey a comprehensive elucidation on TAD hierarchy both in structural definition and in unique functions on gene regulation.

## Results
### Compendium of hierarchical TAD callers
Increasingly more researches are exploring subTAD or the hierarchical structure of TAD, starting from HMM-DI[2] to GRiNCH[41], with over 20

algorithms developed for TAD hierarchy detection (Table 1). There are five principal strategies underlying these tools: linear score, clustering, network features, structural entropy and statistical model (Fig. 1a).

Each strategy represents reasonable interpretation of TAD structures from unique perspective. Linear score shows perspectives for TAD distribution of different sizes by tuning one single parameter, such as Arrowhead by corner score[7], CaTCH by reciprocal insulation (RI)[42], HiTAD[25] by window sizes, OnTAD by size of sliding diamond window and average contact frequency within it[38], Multi-CD[37] by a free parameter $\lambda$ related to domain solution, Armatus[23] and matryoshka[43] by a resolution parameter $\gamma$. The size of TADs is almost positively correlated with the value of the single parameter, and small TADs from low value are usually positioned in large TADs from high value. Thinking TADs as a series of contiguous blocks on a chromosome, clustering iteratively merges TADs neighbors based on similarity of interactions between contact domains to a larger TAD until reaching a chromosome-arm size, and regards the layer-by-layer clustering relationship as the TAD hierarchies, including BHi-Cect[44], SpectralTAD[45], IC-Finder[46], and TADpole[47]. This continuity is also manifested in graph theory, and its two big branches: network features (3DNetMod[48], HBM[49], GRiNCH[41], and spectral[50]) and structural entropy (deDoc[51], and SuperTAD[40]). Network features assume TAD-like structure as the best structural separation on the chromosome, making one TAD as a node and the relation between TADs as an edge. By calculating the edge weight, nodes in the network are divided into vintage clusters, and each cluster covers a large TAD and the nested subTADs within it. The structural entropy is defined over the coding tree of a graph by fixing and decoding the graph in a way that minimizes the uncertainty occurring in random walks. The essence of structural entropy algorithm is to fix the genomic loci at which the uncertainty of the structure is maximized. The less information there is, the more possibility that contact domains are in the same TAD. As for the statistical model, it characterizes TAD hierarchy and biological properties by certain statistical distribution, for example: Gaussian mixture distribution (GMAP[26]), general mixed distribution combined generalized likelihood ratio test (HiCKey[52]), and probability distribution model with dynamic programming (TADtree[53] and PSYCHIC[54]).

Given these methods define the level of TAD hierarchy by different start (from level 0 or level 1) and various directions (from the inside out or the opposite), we make uniform provisions for the level of TAD and boundary (Fig. 1a). TAD is a chromatin structure at the sub megabase scale, which is shown as an isosceles right triangle significantly above the background in Hi-C thermograms. the TAD hierarchical structure is shown as a nested isosceles right triangle (Fig. 1b). To assess how many layers are nested in the TAD structure, we define level 1 for the TADs that don't belong to any larger TAD outside, and the level increases by 1 as smaller TADs position an inner layer. For boundaries, the rule follows that of OnTAD: the maximum of TADs it belonging to in a single direction, suggesting one boundary may belong to different numbers of TADs by left and right (Fig. 1b).

### Evaluation among callers and within each caller
Among the algorithms above, 3D-NetMod requires redundant parameters to be set and tested. For high-resolution data (especially over 40 Kb), there is no precise parameter range, resulting in low confidence for TAD hierarchy prediction; CaTCH seldom gives so proper pair of RIs determining TAD and subTAD that subTADs are not always well located in TADs; results from BHi-Cect are recorded in a complicated form, making it not easy to extract the location of TADs; descriptions for parameters of HMM-DI, HBM, spectral, IC-Finder, PSYCHIC and Multi-CD in each step are not clear enough to perform, even without parameter options. Therefore, these methods do not participate in this evaluation.

After the above screening, we intend to use the Hi-C data available under the Gene Expression Omnibus (GEO)[55] accession number

**Table 1 | Computational methods for prediction of TAD hierarchy**

| Approach | Caller | Input format | Main language | Parameter |
|---|---|---|---|---|
| Liner Score | Arrowhead | hic format | Shell, Awk, Java | 1 |
| | Armatus | dense matrix, sparse matrix, **Rao format**[*] | C++, Python | 1 |
| | CaTCH | catch format[**] | C, R, Shell | 0 |
| | HiTAD | cool format[***] | Python | 1 |
| | matryoshka | dense matrix, sparse matrix, **Rao format** | C++, Shell | 1 |
| | OnTAD | **dense matrix**, hic format | C++ | 2 |
| | Multi-CD | dense matrix | Matlab | NA |
| Clustering | IC-Finder | dense matrix, sparse matrix | Matlab | NA |
| | TADpole | dense matrix | R | 6 |
| | BHi-Cect | Rao format | R | 0 |
| | SpectralTAD | dense matrix, sparse matrix, hic format, cool format, **Rao format** | R | 3 |
| Network features | HBM | dense matrix | R | 5 |
| | spectral | mat format[****] | Matlab | NA |
| | 3DNetMod | sparse matrix | Python | 18 |
| | GRiNCH | sparse matrix | C, Python | 3 |
| Structural Entropy | deDoc | sparse matrix | Java | 0 |
| | SuperTAD | **dense matrix**, sparse matrix | C++ | 0 |
| Statistical Model | TADtree | dense matrix | Python | 6 |
| | GMAP | dense matrix, **sparse matrix** | R | 4 |
| | PSYCHIC | dense matrix | Matlab, C | NA |
| | HiCKey | dense matrix, sparse matrix, **Rao format** | C++ | 6 |

A matrix is a two-dimensional data object made of m rows and n columns, therefore having total m x n values. If most of the elements of the matrix have 0 value, then it is called a sparse matrix. In Hi-C data, the sparse matrix represents the chromatin contact map, the numerical values in row i and column j represent the frequency of DNA interaction between i bin and j bin in chromosomes. The sparse matrix is one of the common inputs for TAD hierarchical structure recognition algorithms.
[*]Rao format is another sparse matrix, of which the start and end sites are represented by genomic coordinate.
[**]catch format, [***]cool format, and [****]mat format mean files separately produced by CaTCH, cooler, and Matlab.
Bold text is the actual input type for this article.

GSE63525 to compare performance of the following 13 methods: Arrowhead, Armatus, TADtree, HiTAD, GMAP, deDoc, matryoshka, OnTAD, TADpole, SpectralTAD, HiCKey, SuperTAD and GRiNCH. According to length range of TADs/subTADs, we pick data resolutions including 5 Kb, 10 Kb, 25 Kb, 50 Kb and 100 Kb on chromosome 7 from 7 *human* cell lines (GM12878, IMR90, HMEC, HUVEC, NHEK, K562, KBM7), and chose MAPQ > 30 data normalized with the iterative correction and eigenvector decomposition (ICE)[56] (see the Methods), under certain condition. Chromosome 7 contains moderate genetic messages that ensure the reliability of the analysis, and covers vital genes like the *HOXA* gene family, which are involved with genome architecture, limb development, and multiple types of cancer[28,57–59].

**Variation of hierarchical TAD structures from different algorithms.** Short of the metric to compare such domain-within-domain structure, we developed hierarchy structural similarity (Hier_SSIM) (see the Methods) using structural similarity to judge the similarity of the output heatmap (Fig. 2a).

We obtain the output results from all the tools on 10 Kb data of GM12878 cell line (Fig. 2b, c). We categorize these methods by average linkage (see the "Methods" section), a kind of hierarchical clustering, and obtain four main clusters: the first includes HiCKey and SuperTAD; the second involves only GRiNCH; the third contains matryoshka, SpectralTAD, Arrowhead and deDoc; the last covers Armatus, OnTAD, GMAP and HiTAD (Fig. 2b). Overall, there is a high degree of similarity between all clusters. Though such clusters are not significantly consistent with methods groups, the diagram of TAD hierarchy directly exhibits two possible reasons. For example, TADs located at 1–1.5 Mb have the following conditions: (1) belong to the upstream or the downstream larger TAD; (2) consist of scattered monolayers or hierarchical construction (Fig. 2c). Disagreement of the four clusters above partly consults from the division at a larger scale (larger TADs),

confirming that TAD confines the nested subTADs within it whatever their reconstruction[36]. That explains the difference within the cluster. While the hierarchy in the same genome region has greater power to distinct clusters. Hence, TAD hierarchy could prove a decisive factor in chromosome topology and deserves further concern.

Since lack of a conclusive gold standard to evaluate the accuracy of TAD prediction, it's a feasible option to judge the correlation with biological features. Given that TAD boundaries are always anchored by architectural proteins such as CTCF and cohesion[2,7–9], we compared the enrichment of CTCF and SMC3 (subunit of cohesin protein complex) at TAD boundaries from all tools. Both markers show sharp signal peak on OnTAD, HiTAD, Armatus, deDoc, matryoshka, Arrowhead, and SpectralTAD. GMAP and GRiNCH show good correlation with CTCF signal peak around boundaries (Fig. 2d, Supplementary Fig. 1a), reflecting perfect accuracy and efficiency in detecting TAD boundaries and best recognition of TAD segregation. To verify the robustness of our results across species, we applied TAD hierarchy recognition methods to Hi-C data of *mouse* CH12-LX cell line and *drosophila* S2 cell line. Notably, we consistently found that CTCF enrichment occurred at TAD boundaries in different species and achieved the most enrichment in the OnTAD boundaries (Supplementary Fig. 1b, c). Apart from CTCF and cohesin, TAD boundaries tend to enrich active regulators, like H3K4me3 (active promoter), H3K27ac (active enhancer and promoter), POLR2A (subunit of transcriptional factor) and H3K9ac (active promoter) but don't collect repressed elements such as H3K27me3 (repressed promoter) and H3K9me3 (heterochromatin)[2,35]. We conduct the same analysis on these markers: DNase-seq, H3K4me3, H3K27ac, POLR2A, and H3K9ac. These markers permanently show significant signal peak around boundaries from OnTAD and Armatus and occasionally show it on HiTAD, matryoshka, Arrowhead and SpectralTAD (Supplementary Figs. 1d, e, and 2). And HiCKey stably shows little but clear peak of all the above markers (Fig. 2d,

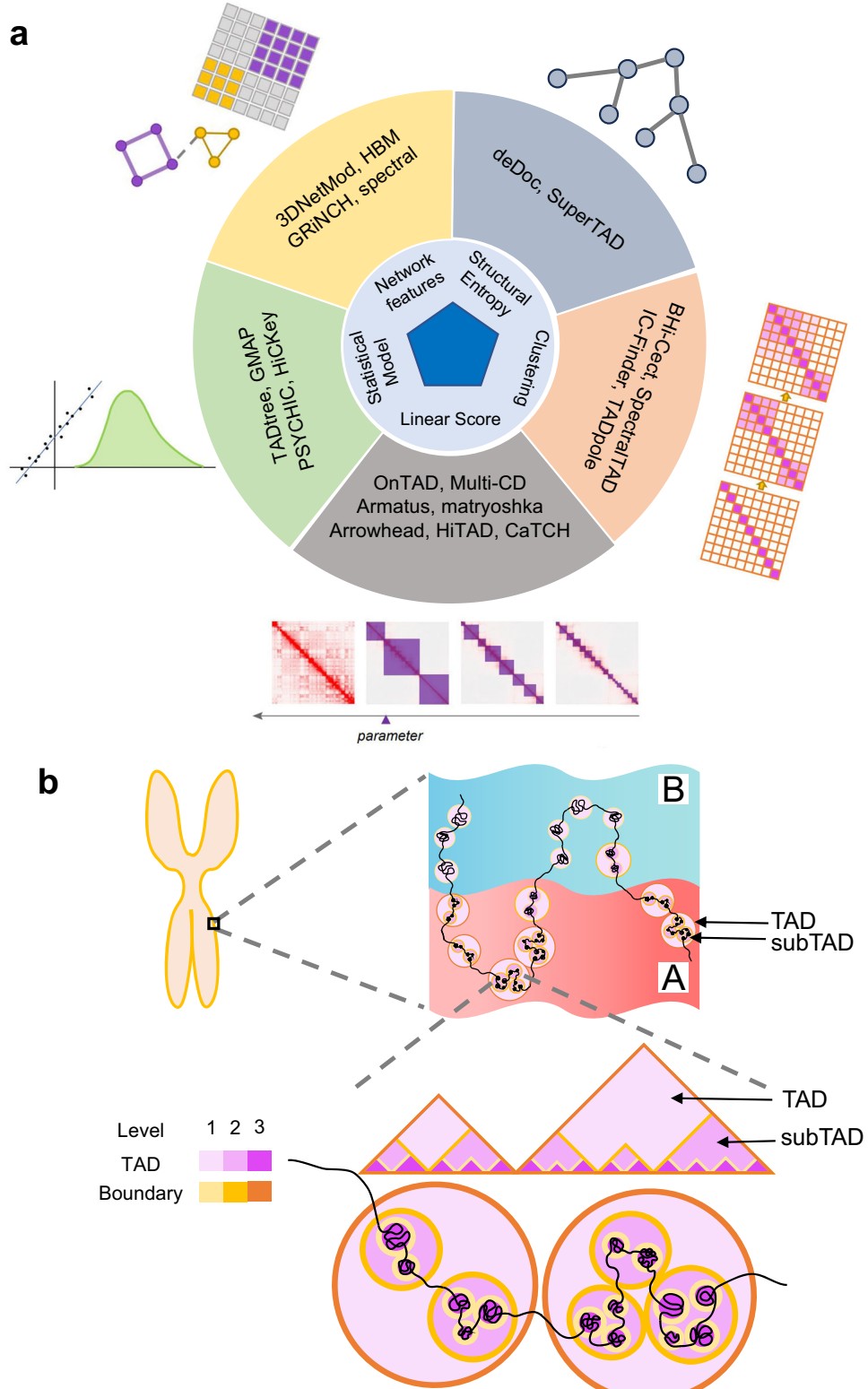

**Fig. 1 | Compendium of TAD hierarchy and callers. a** Categories of TAD hierarchy callers. **b** Definition and calculation of TAD hierarchy. Top: Structure of TAD at chromosome (the left panel) and A/B compartment scales (the right panel), red area indicates A compartment, blue area indicates B compartment. Bead-like structure refers to TAD on the genome. Middle: spatial morphology of TAD and subTAD in the nucleus, light purple represents TAD, dark purple represents subTAD, dark yellow represents TAD boundary, light yellow represents subTAD boundary. Lower: TAD hierarchical structure in Hi-C heatmap, the color is consistent with the middle figure. The darker purple color represents higher TAD layers. Darker yellow represents higher TAD boundary hierarchy.

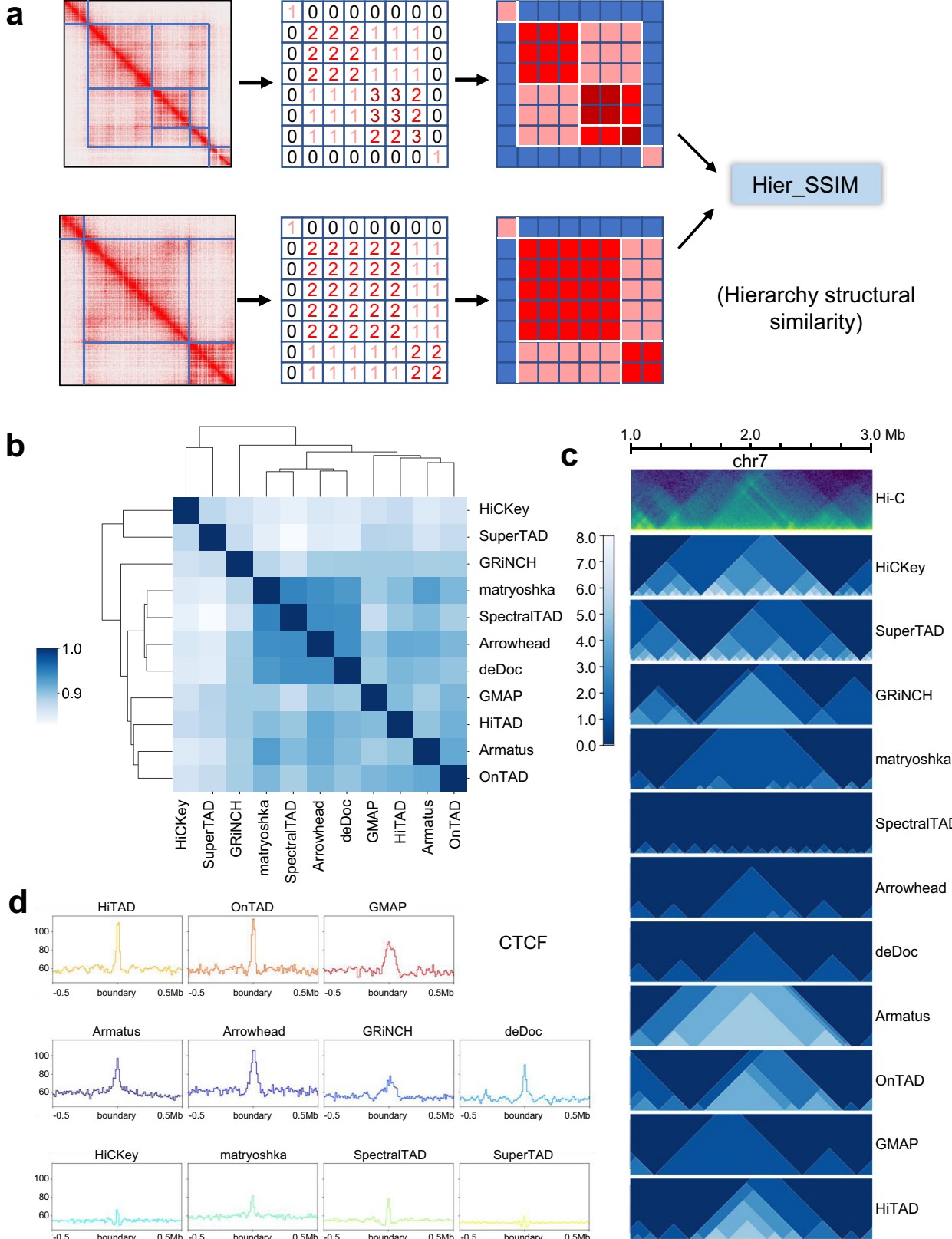

**Fig. 2 | Hier_SSIM and evaluation across all callers (ICE-normalized Hi-C data at 10Kb resolution). a** Diagram of Hier_SSIM process. **b** Clustering of TAD hierarchy callers. Source data are provided as a Source Data file. **c** Hierarchical structures of representative region by callers. The green image in the first row represents the Hi-C heatmap. The remaining blue images show the distribution of TAD levels from each method. **d** Peak signals for structural protein CTCF around the boundary.

Supplementary Figs. 1 and 2). While H3K27me3 is apparent on Armatus and matryoshka, H3K9me3 peak is clear on GRiNCH, deDoc, and GMAP (Supplementary Fig. 3a and b). Moreover, we search for overlap of boundaries with protein-coding genes, including promoter and coding sequence (CDS), and consistently find boundaries from SuperTAD cover the greatest number of genes, followed by HiCKey, matryoshka and SpectralTAD (Supplementary Fig. 3c and d). Together, we conclude that OnTAD, HiTAD, SpectralTAD, and Arrowhead are perfect for detecting architectural proteins and active epigenomic indicators. GRiNCH, deDoc, and GMAP perform well in capturing inactive regulators and architectural proteins. Armatus and matryoshka do good in all three aspects.

**Hierarchical TAD identification across data resolution, normalization, sequencing depth and biological replicates.** For some basic features, we first evaluate these methods with ICE Hi-C data in TAD segment length, number/percentage of TAD/boundary at all levels, and the genomic coverage of TADs. Researches generally consider the length of TAD and subTAD roughly in the range of 30 kb-2 Mb, so we confine the size of TAD segments within this limit for subsequent analysis and comparison (see the Methods), except for comparison of TAD segment length.

We first explore the length distribution of TAD segments on 7 cell types and all five resolutions without size filtration on the ICE Hi-C data (Fig. 3a, Supplementary Fig. 4). Except Arrowhead and TADpole, the length of TAD fragments is generally within 2 Mb, of which is within 1 Mb in deDoc, matryoshka and SuperTAD. There is a trend of length shortening with the lifting of resolution in most of the methods, while that of GRiNCH fluctuates in a stable range (Fig. 3a, Supplementary Fig. 4).

Then we compare the number and percentage respectively of TADs and their boundaries of all levels on the 50 Kb and 10 Kb ICE Hi-C data of GM12878 and K562 (Fig. 3b, c, Supplementary Fig. 5), and we find a relatively consistent distribution in most approaches (except SuperTAD and HiCKey). The number and percentage of TAD segments decrease as the level arises, with the same tendency seen in TAD boundaries (Fig. 3b, Supplementary Fig. 5a, c, d). With the increase of resolution, the number of TAD rises at all levels, especially the higher level in Armatus, OnTAD, HiTAD, and GMAP, as well the proportion (Fig. 3b, Supplementary Fig. 5a, c, d). As for TAD boundaries, the increase in resolution makes the number of boundaries measured by all methods increases in each level, but the ratio between the levels remains permanent (Fig. 3c, Supplementary Fig. 5b, e, f).

Next, we define the percentage of TAD segment coverage on the whole chromosome as the genomic coverage and count it on chromosome 7 (see the "Methods" section). The genomic coverage of most methods is over 90% (Fig. 3d, Supplementary Data 1). For OnTAD and TADtree, the values are stable at 80% and 70%, respectively, while the ratios of SuperTAD, HiCKey, and TADpole are almost up to 100%. In general, the genomic coverages of the majority of tools are slightly affected by resolution, while those of matryoshka and Arrowhead are largely affected by resolution.

Robustness is an important metric to evaluate performance of TAD hierarchy callers. We set a series of testing conditions by changing resolution (5 Kb, 10 Kb, 25 Kb, 50 Kb, and 100 Kb), matrices normalization (raw matrix and ICE-normalized matrix), and sequencing depth (20%, 50%, and 100%) (Fig. 4). Here, two metrics are used to measure the similarity of TAD hierarchy: overlap ratio (OR, derived from SuperTAD[40] to evaluate the similarity between two coding trees) and Hier_SSIM (see the Methods).

To objectively assess the performance, we set similarity of 0.7 as a standard, which conveys a neutral correlation. Our Hier_SSIM results reveal that matryoshka, HiTAD, TADpole, deDoc, Armatus, SuperTAD, OnTAD, Arrowhead, TADtree, GMAP and SpectralTAD are less affected by resolution (Fig. 4a), while OR shows good robustness of deDoc,

GRiNCH, OnTAD, SuperTAD, and TADpole (Supplementary Table 1). As for raw data and ICE-normalized data, SpectralTAD, Arrowhead, SuperTAD, deDoc, and TADpole show high similarity, indicating that they are superior in processing raw data (Fig. 4b, c). Finally, after downsampling the ICE-normalized data of 50 Kb by 50% and 20%, we find that SpectralTAD, matryoshka, deDoc, TADpole, OnTAD, GMAP, TADtree, GRiNCH, and SuperTAD are seldom affected by the sequencing depth of input data (Fig. 4d, Supplementary Table 2). Owing to the diversity in sequencing depths of 7 cell types (GM12878, HMEC, HUVEC, K562, KBM7, and NHEK) in GSE63525 dataset, we apply OnTAD on 50 Kb and 10 Kb ICE-normalized data and calculate OR between every two samples to seek the leading factors of TAD hierarchy variation. Results are mainly divided into two distinct clusters by resolution, with OR ranges from 0.7 to 0.8 within the same cluster and that around 0.5 between distinct clusters (Fig. 4e, Supplementary Fig. 6a). Sequencing depth and cell-specificity may give rise to the fluctuation within the cluster. Together, data resolution serves as the principal component of discrepancy among results from one single algorithm. Lastly, to measure the reproducibility of TAD hierarchy calling results on biological replicates of Hi-C data, we applied TAD hierarchy recognition methods to GM12878 Hi-C data, and found that matryoshka, SpectralTAD, OnTAD, Arrowhead, GMAP, and HiCKey can achieve great reproducible results (Supplementary Fig. 6b). And SpectralTAD can achieve the most reproducible results with 0.976539 similarity. This suggested that these methods are robust in identifying TAD in biological replicates.

In summary, we find that TAD hierarchy is greatly affected by resolution and sequencing depth, but seldom varies with normalization and biological replicates. While there are still plenty of hierarchical TAD callers that show stability whatever the resolution, with similarity ranging around 0.7. As for traditional TAD callers, the similarity between multi-resolution is about 0.5[24]. That means TAD hierarchy is less influenced by resolution, compared to single TAD structures. Thus, focusing on TAD hierarchy is promising to overcome the shortage or limitation on resolution and convey potential information of chromatin, which would be a wiser choice than single TAD.

**Comprehensive performance and guidance of hierarchical TAD callers**

Based on the testing above, we summarize a comprehensive evaluation of all methods (Fig. 5a), including biological correlation, robustness, and actual user experience (software installation, code instructions, input processing, parameters setting, downstream procession, time consumed, resolution self-identification and built-in visualization). We also provide details of running time and memory cost for each of the 13 methods (Supplementary Table 3). From the perspective of biological correlation, OnTAD and Armatus perform extraordinarily in architecture proteins, chromatin accessibility, and active histone modifications, while Armatus, matryoshka, GRiNCH, deDoc, and GMAP are suitable for inactive molecular markers. As for input, spectral requires matrix from Matlab; CaTCH needs catch file produced by hicpro2-catch; the cool matrix in HiTAD could be obtained by HiCExplorer; deDoc deals with sparse matrix with max bin order in the first line; TADtree, HiCKey, HiTAD and 3DNetMod conduct configure files containing parameters, the path of input and software. Such variety can be seen in output: OnTAD, HiTAD, matryoshka, and GMAP directly show coordinate and level of each TAD; deDoc, SpectralTAD, and GRiNCH just send TADs at different levels in separate files; the rest of the methods require the calculation of TAD level additionally. This evaluation is not totally equal to the real performance of tools, affected by the quality of library construction, sequencing depth, and background noise. The input of single-cell Hi-C data provides a more realistic three-dimensional structure of chromatin, but at the same time poses the problem of high noise and data sparsity. In addition, disagreement of the results partly originates from various strategies or principles of

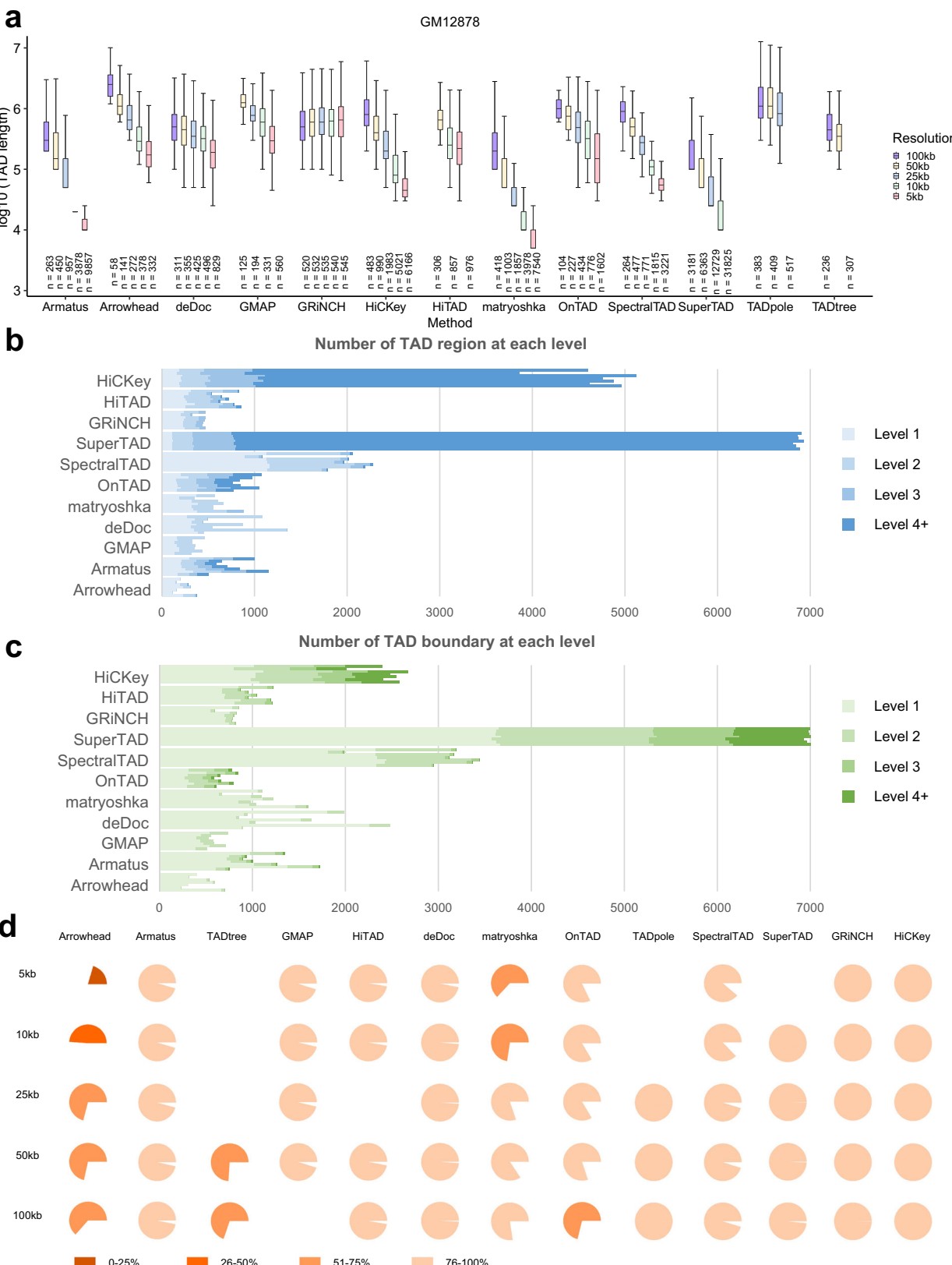

Fig. 3 | Identification of hierarchical TADs in chromosome 7 from ICE-normalized Hi-C data. a Length range of hierarchical TADs called by different methods and resolutions. The line that divides the box into 2 parts represents the median of the data. The ends of the box show the upper (Q3) and lower (Q1) quartiles. The difference between Quartiles 1 and 3 is called the interquartile range (IQR). The extreme line shows Q3 + 1.5xIQR to Q1-1.5xIQR (the highest and lowest value excluding outliers). b, c Numbers of TADs (b) and boundaries (c) at various levels of the GM12878 cell line on 10 Kb. d Genomic coverage of hierarchical TADs of various callers and resolutions. Source data are provided as a Source Data file.

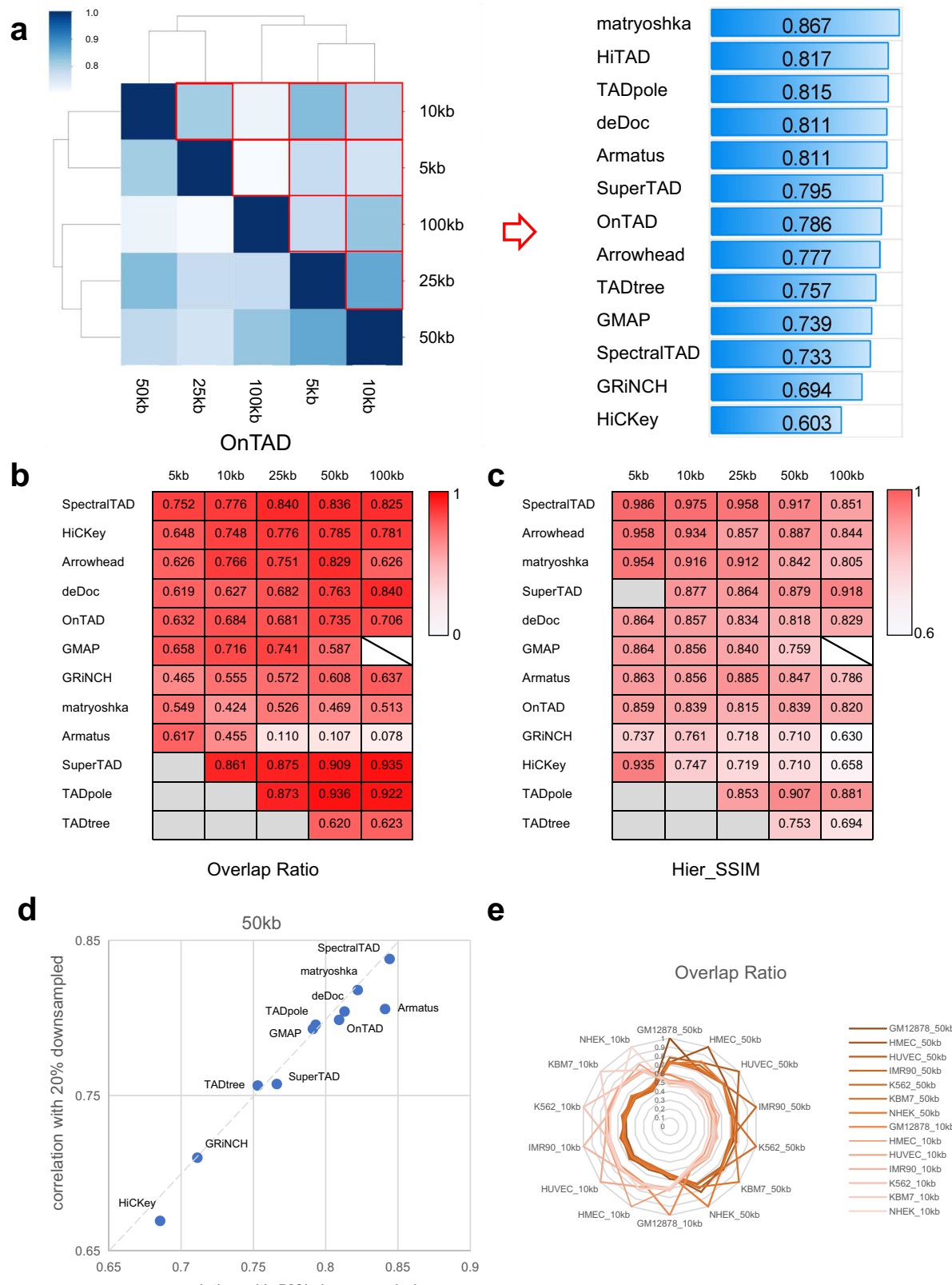

TAD prediction, showing diverse understandings of TAD structure and hierarchy. Together, we provide a commendation of TAD hierarchy callers with proper input format for colleagues interested in it from fields of chromatin 3D structure, life science, and medicine (Fig. 5).

Besides, it's important to flexibly select the appropriate method according to the resolution of sample data, sequencing depth or data sparsity, platform sources, and sequencing technologies. First, what should be identified include the sequencing depth (or data sparseness) of data and the optimal range of resolution. When dealing with higher-resolution data, OnTAD, HiTAD, matryoshka are ideal options, and matryoshka also performs well on low-resolution data (-500 kb)[43]. In terms of sequencing depth, SpectralTAD, deDoc, and GRiNCH can handle ultra-sparse data. It is worth mentioning that IC-Finder and

**Fig. 4 | Evaluation of each caller across data resolution, normalization, and sequencing depth. a** Hier_SSIM between TAD hierarchy obtained at different resolutions was assessed in a pairwise manner (e.g., 5 Kb vs. 10 Kb, 5 Kb vs. 25 Kb, etc.; results for the ICE data only are shown here). Hier_SSIM varies from 0 (no similarity, white) to 1 (full similarity, dark blue), showed in the Heatmap simulation diagram (the left panel). TAD hierarchy callers are ranked based on the average values of the Hier_SSIM across all resolutions (from highest to lowest, the right panel). **b, c** Concordance between TAD hierarchy obtained with each caller from raw and ICE-normalized matrices at different resolutions (5, 10, 25, 50, 100 Kb)

using the Overlap ratio (**b**) and the Hier_SSIM (**c**). Overlap ratio and Hier_SSIM vary from 0 (no similarity, white) to 1 (full similarity, dark red). TAD hierarchy callers are ranked based on the average values of the overlap ratio and the Hier_SSIM (from highest to lowest). Samples are beyond the resolution range of callers (backslash) and results of certain resolutions can not computed (gray). **d** Ratio of Hier_SSIM of TAD hierarchy between 20% and 100% versus that between 50% and 100% from GM12878 cell line 50 Kb ICE data. The dashed line indicates the linear fit. **e** Overlap ratio between TAD hierarchy obtained with 7 cell lines on 50 Kb ICE data by OnTAD. Source data are provided as a Source Data file.

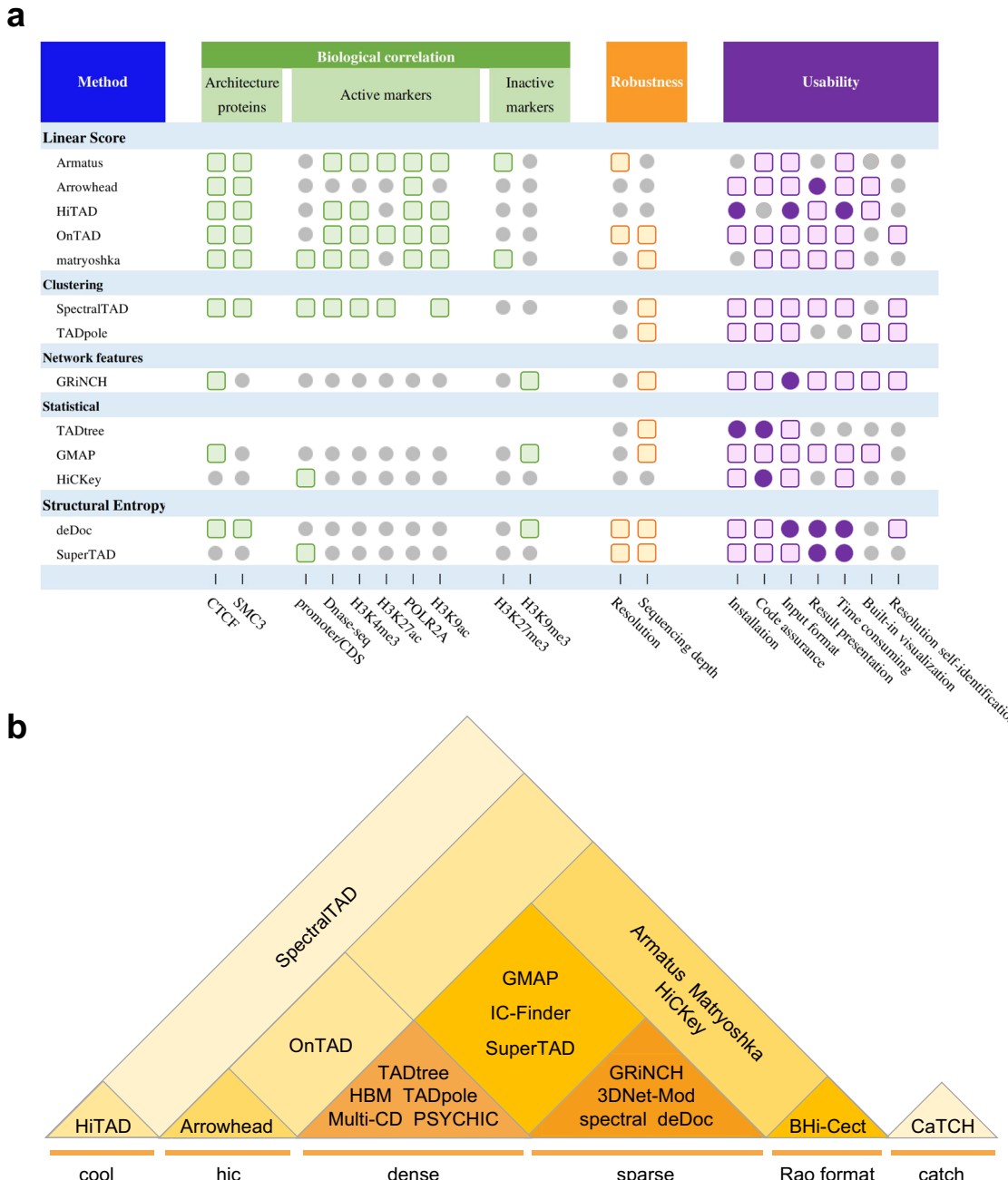

**Fig. 5 | Comprehensive evaluation of TAD hierarchy callers. a** Summary performance of methods. The grey circles, colored circles, and colored squares represent normal, good, and excellent respectively. **b** Input format required by methods.

SpectralTAD work well in high-noise inputs. Second, for samples from various sequencing technologies and platforms, the following methods have their special focus. All methods are capable of processing bulk Hi-C data, while Armatus and chromoHBM-3C additionally deal

perfectly with 3 C data; deDoc is friendly to ultra-sparse data and even pooled single-cell Hi-C data; TADpole is suitable for Capture Hi-C data after combing DiffT scores; PSYCHIC shows compatibility for data from multiple platforms, such as SPRITE, HiChIP, and Hi-C.

Since performance related to the biological significance of all methods shows no huge difference, we summarize a guidance more focused on usability (Supplementary Fig. 7). The guidance takes three main points into consideration: (1) complexity of format for input/output files; (2) computational memory and resource consumption required by software; (3) abundance of the TAD hierarchy obtained. OnTAD shows predominant smooth in format resolvent of files, abundant levels of TAD hierarchy, and little consumption in running. GMAP is user-friendly while presenting steady two layers. If the computing platform is strong, SpectralTAD, Armatus, matryoshka, and deDoc are wise choices. Researchers who are skilled in file format conversion can try deDoc, Arrowhead, matryoshka, and HiTAD.

## Applications of TAD hierarchy in biomedicine

Studies to date using these methods mainly focus on the formation of chromatin construction, gene regulation, embryonic development, and disease. First, the mechanism of structural formation has been explored throughout the field. OnTAD is used to describe the topological structure in normal and H1 depletion T cells, finding the impact of histone H1 in chromatin compaction[60], and capturing chromosome conformation of *human* sister chromatids[61]. It also shows distinct TAD hierarchy in normal colon and colorectal carcinoma[39]. HMM-DI is applied to identify a subset of TADs exhibiting strong core-periphery mesoscale network in T cell[62]. Armatus provides evidence that TAD cliques are general phenomena across kinds of cell types and reveals genomic differences between large TAD cliques and small ones[63]. deDoc, HiTAD, and GMAP respectively detect TAD construction in pig embryos[64], maize[65], and *mouse* erythroid cell populations[36]. Moreover, the spatial scale of chromatin architecture is vital for tool selection: Multi-CD, GRiNCH, and CaTCH can deal with TADs, sub-TADs, and compartments, while chromoHBM (one of the algorithms of HBM) and PSYCHIC perform well in loop interactions, thus especially suitable for studying interactions between regulatory elements and target genes.

Second, the relationship between chromatin topology and genome function is still contentious. And some methods have made an attempt at gene dysregulation. Two clear TADs detected by SpectralTAD are mapped with numerous gene-gene and gene-enhancer interactions around the Regulators of Complement Activation (RCA) gene cluster in B cells, revealing extensive co-regulation[66]. deDoc is used in heat shock (HS) and calls TADs in the K562 cell line under normal HS (NHS), short-term HS (SHS), and long-term HS (LHS). And it helps researchers find strong stability of chromatin in response to SHS, compared to little alteration of chromatin accessibility[67].

Third, a few methods have generated progression in embryonic development. HiCKey, Arrowhead, OnTAD, SpectralTAD, and TADpole are used to confirm insulation ranking of germ cells at various stages and lineages of differentiation, facilitating the principle for the nucleus programming that creates gametogenic progenitors in both sexes[68]. 3DNetMod, together with allelic expression states and chromatin marks, is used to explore the alteration of chromosome topology among oocytes, sperm, and early preimplantation embryos, exploring the complex dynamics of 3D-genome organization during early development[69].

Last but not least, researches on chromosome spatial structure have been centered around disease progression. HMM-DI reveals that the linkage disequilibrium (LD) blocks encompassing schizophrenia disease-associated single nucleotide variants (daSNVs) are significantly enriched at core nodes, whereas obsessive-compulsive disorder (OCD) daSNVs are enriched at periphery nodes and autism spectrum disorder (ASD) daSNVs are equally distributed across core or periphery nodes, indicating link between 3D-genome's core-periphery network structure and neuropsychiatric daSNVs[62]. OnTAD can distinguish normal colon from colorectal carcinoma both with hierarchical level of TAD

and involved genes[39]. Chromosome spatial architecture, identified by SpectralTAD, shows the downstream efforts of loss of NSD1 in head and neck squamous cell carcinoma (HNSCC), together with RNA-seq results. That presents genome structure's proficiency in targeted treatments[70]. 3DNetMod identifies subTADs in ESCs to help discover the relationship between high-scale genome topology alteration and disease-associated short tandem repeats (STRs), which is well known to contribute to over 25 inherited disorders[71]. In addition, TADpole performs perfectly in insertion points of Inv1 mutations[47]. Thus, it is necessary to say that some methods are suitable for specific biological problems.

## The impact of cell heterogeneity on TAD hierarchy

As for bulk Hi-C data, TAD levels are consistent with the extent of gene expression, but the formation and existence of TAD hierarchy require further studies. Currently, the hypothesis for the formation of TAD hierarchy is divided into three categories: one is that the TAD hierarchy exists in a single-cell adjusting the gene expression (the pink box); the second regards it as just single superimpose of TAD layers derived from millions of cells (the blue box); the third supports concurrence of the previous two points (the orange box) (Fig. 6a). In clinical samples, the authenticity of TAD hierarchy greatly affects the study on triggers of disease. Our recent work proposed that the heterogeneity of cancer cells may be one of the reasons for the formation of TAD multi-scale layers in complicated TAD hierarchy in colorectal carcinoma[39].

To explore the impact of cell heterogeneity on TAD hierarchy, we perform a simulation of cellular heterogeneity by mixing GM12878 and K562 in 11 different ratios (Supplementary Fig. 8a, see the Methods). Given the evaluation above, we observe that OnTAD exhibits better performance in identifying TAD hierarchy, thus it enables us to provide convincing results for subsequent studies. OnTAD is used on the mixed data, and to search changes in TAD/boundary levels. If TAD hierarchy is the result of image superposition, the number of TAD boundaries with high levels will get obviously higher at a certain mixing ratio than that of a pure cell line. Interestingly, we find that the number doesn't significantly excess that of K562 regardless of the mixing ratio (Fig. 6b). Further, we add IMR90 to be mixed with GM12878 and K562 in equal proportion, compare them with cases of their pure cell lines, and obtain consistent results (Supplementary Fig. 8c). As mixing ratio changes, the number of TADs tends to be approximate to that of the pure cell line which has more TADs. A similar phenomenon can be seen for boundaries (Supplementary Fig. 8b, c).

Next, to better simulate the impact of cell heterogeneity on TAD hierarchy, we collected single-cell Hi-C data of GM12878 and IMR90 from Kim et al.[72]. Then, we generated pseudo-bulk Hi-C matrices based on different cell mixing ratios and calculated the distribution of TAD numbers at different levels (Supplementary Fig. 8d). The results were consistent with bulk Hi-C mixing. We noticed the outputs never excess greatly to one pure cell line whatever the mixing ratio. In this way, the distribution of TAD hierarchy is seldom affected by cellular heterogeneity. Hence, we infer that TAD hierarchy would never just be a superimpose of millions of Hi-C heatmaps. This suggested that TAD hierarchy could be a real architecture in single cells.

Our discovery is consistent with recent findings. Xiaowei Zhuang group conducted a super-resolution chromatin tracing method based on multiplexed stochastic optical reconstruction microscopy (STORM) imaging of chromatin architecture and discovered TAD-like domains with spatially segregated globular conformations in single cells, of which the boundary shows cell-to-cell variety[73]. Besides, Jian Ma Lab and Zhihua Zhang group separately developed Higashi[74] and DeTOKI[75] to identify TAD-like domains with scHi-C data. Now that the existence of TAD-like domains has been confirmed with imaging technology, the discovery of TAD hierarchy at the single-cell level is just around the corner and has great research potential.

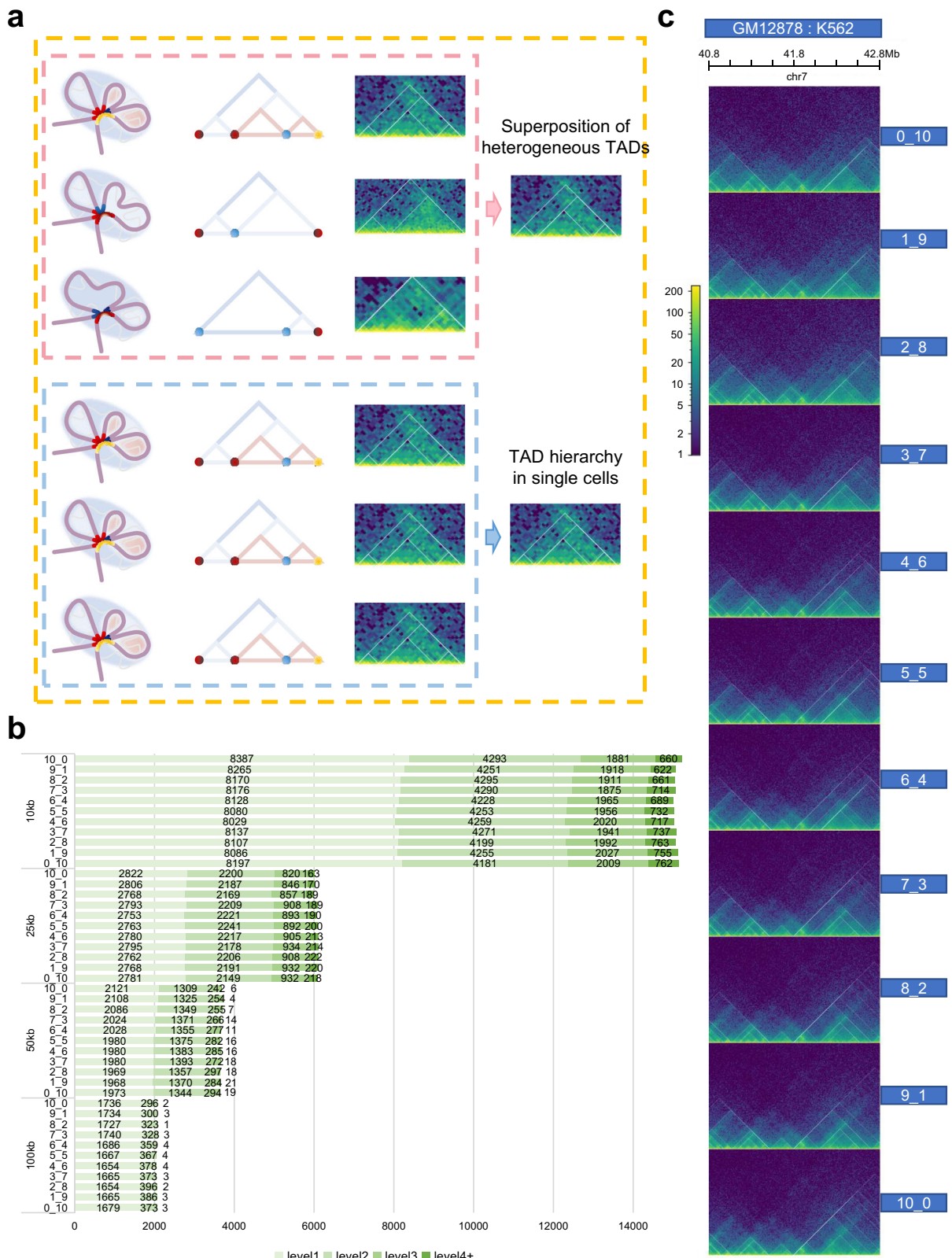

**Fig. 6 | Exploration for existence of TAD hierarchy. a** Schematic diagram of three main hypotheses for the formation of TAD hierarchy. One is only the superposition of heterogeneous TADs from bulk data (the upper panel, pink), the second shows that exact TAD hierarchy exists in individual cell (the lower panel, blue), and the third supports the coexistence of the two (orange). Twisted light purple chromatin filaments form the TAD, highlighted by light blue circular shading. Various colored points or short curve located at convergence points indicate TAD boundaries (the left panel, pink and blue box). Hierarchical TAD pattern diagrams (the middle panel, pink and blue box). Actual hic heatmap (the right panel, pink and blue box). **b** Number of TAD boundaries at separate levels and various resolutions in mixed samples. Source data are provided as a Source Data file. **c** TAD hierarchy in 40.8-42.8 Mb on chr7 of all mixed samples.

## Hierarchical TADs act as air conditioner

Previous studies have pointed out that TAD boundaries are formed by cohesin protein complex sliding on chromatin fibers until encountering convergent CTCF[10], nominated as 'loop extrusion' model[76]. Thus, TAD and subTAD can be described as 'loop domains'[77]. Beyond this, the subTAD boundary generates insulating neighbors featured by significantly distinct genome function, and the 'asymmetric ring extrusion' model is proposed to explain the mechanism of TAD hierarchy, among which the boundary with levels 5+ obsesses significantly high transcriptional activity[38].

In the section of the evaluation, OnTAD performs well in the majority of aspects, thus we use it to conduct deep analysis on ICE-normalized data of GM12878 (10 Kb). We found that the enrichment of H3K4me1 and H3K4me3 (related to the active promoter) increased as the TAD and boundary levels went up (the yellow box), while that of H3K27me3 (associated with the repressed promoter) was the opposite (Fig. 7a, Supplementary Fig. 9a). This is consistent with previous findings of TAD hierarchy[38]. In addition, we predict the hierarchical TAD structure of GM12878 and K562 cell lines, and compare the gene expression along with epigenetic features (Fig. 7b, Supplementary Fig. 9b). It is obvious that in the 86–88 Mb region of chromosome 7, the TAD hierarchy is rich in GM12878, while barren in K562, which coincides with the epigenetic landscape and expression abundance. It also confirmed the previous finding that hierarchical TADs are more active than single TAD or TAD gaps[38]. Moreover, both cell lines have the TAD encompassing the cyclin D binding myb-like transcription factor 1 gene (*DMTF1*) (86.79–86.87 Mb, the yellow shading). What the two cell lines have in common is that the TAD border is level 3 at 87.86 Mb (the pink shading). But the TAD border in GM12878 shows the interaction with the *DMTF1* TAD, while failing to link with the *DMTF1* TAD in K562. Moreover, the *DMTF1* TAD is encompassed by the larger TAD (86.88–87.85 Mb) in GM12878 with more plentiful hierarchy, but that of K562 is distinct. These two points may result in the contrast expression of *DMTF1*. Thus, higher-level boundaries of the *DMTF1* TAD will affect the expression of *DMTF1*.

Based on this, we propose an air conditioner model for the mechanism of TAD hierarchy (Fig. 7c, Supplementary Fig. 9c): The air conditioners represent enhancers. Then various levels of TADs and TAD boundaries are represented by shadows and curves of different colors. As shown in Fig. 7c, the high-level TAD boundary mediates the formation of three TADs. As a result, high-level TAD boundary has high interaction frequency with other TAD boundaries. Since active modifications and gene transcription were enriched at the TAD boundary, we inferred that there were active enhancers and transcriptional-activated genes on the TAD boundaries. A high-level TAD boundary can gather a number of enhancers by interacting with other TAD boundaries. Enhancers play the role in activating transcription by near-space interaction, so there is an analogy between enhancer and air conditioner. The more concentrated the distribution of enhancers, the stronger the activation of genes. In different cells types, states or conditions, TAD hierarchy might change. The reduction of TAD boundary level leads to reduction of enhancer concentration (Supplementary Fig 9c). And the reduction of enhancer concentration limits the activation of gene expression. This is consistent with our model that the concentration of air conditioner affects the ability of controlling the temperature of the aggregation area.

Hence, in GM12878, the air conditioner concentrated in the high-level TAD boundary, indicating *DMTF1* TAD boundary has highly active modification (Fig. 7b, Supplementary Fig. 9b, yellow shadowing). Thus, the expression of *DMTF1* is high with the help of highly concentrated air conditioners. While in K562, the reduction of the *DMTF1* TAD boundary level may lead to lower concentration of enhancers, which limits *DMTF1* gene expression. Same result can be also seen for ADAM metallopeptidase domain 22 gene (*ADAM22*) (the pink shading). DMTF1 protein helps to suppress cell growth or induce apoptosis, and

*ADAM22* encodes protein without metalloprotease activity. Lower expression of these genes in K562 may be propitious to tumorigenesis. To further exploration, we analyzed the TAD hierarchy in colorectal carcinoma (BRD3187) and paracancerous tissue (BRD3187N), together with gene expression (Supplementary Fig. 10a). Interestingly, the boundary of TAD containing the acylglycerol kinase (AGK) gene in colon cancer tissues was more spatially folded, resulting in an increased density of active regulatory elements in the surrounding region. Transcription level of *AGK* was also elevated. While in paracancerous tissues, the reduction of *AGK* TAD boundary level made the transcriptional active elements sparser in the adjacent space, contributing to lower AGK expression. The *AGK* gene encodes a mitochondrial membrane protein involved in lipid/glycerolipid metabolism and oncogenic MAPK signaling, and its higher expression in colorectal cancer tissues also promotes cancer development. Generally, TAD hierarchy shows more complexity in genome regions with higher transcriptional activity.

The air conditioner phenomenon can also be observed in TAD hierarchy between K562 and IMR90 in previous study[52]. The spatial folding degree of chromatin in region anchored by CTCF caused the difference in density of transcription active elements, resulting in different transcription levels of genes at the same location between different cell types, states, or developmental stages. The degree of folding resulted in the spatial proximity of genomic fragments, and this spatial proximity does not have the same high insulation fraction as the CTCF anchor, allowing greater flexibility in the arrangement of subTADs within a large TAD. This is consistent with the inter-cellular heterogeneity of subTADs. In sum, the high-level TAD boundary indicates the convergence of numerous powerful regulatory factors like super-enhancers and is functionally similar to 'hub-boundaries'[38]. Together, the air conditioner model will provide a broader and more flexible perspective to explore genes, transcription factors, enhancers, histone modifications, and their interaction.

## Discussion

Genome regulation is promising to be more precise at subTAD level, and TAD hierarchy plays a critical role in ontogeny and diseases. We make systematic definition and grades of TAD hierarchy. With five main types of tools accessible, we compare TAD hierarchy from 13 hierarchical TAD callers with a metric Hier_SSIM, from the perspective of biological correlation, robustness under raw/ICE-normalized data, multiple resolutions, and various sequencing depths. We find TADs and boundaries of all levels variable mostly by resolution and method. Integrating the above factors, a comprehensive evaluation and operational user manual are present for TAD hierarchy callers. To a large extent, our work deepens understanding of TAD hierarchy and recommends tools for the best match, ensuring stability of results and perfect performance. Thus, hierarchical approaches are promising to promote more biological applications in biomedicine.

Existence of TAD hierarchy in individual cells may reveal instantaneous scenes of gene activity making up the complete process in certain cell type or state and partly explain the disagreement in the previous comparison of 22 TAD callers[24]. Thus, the focus is on the alteration of hierarchical TAD layers, replacing the single difference in TAD size or number. Besides, TAD hierarchy is confirmed to change greatly between normal colon and colorectal cancer[39], which could be a breakthrough of involved researches in medical fields. In this way, TAD hierarchy is hopeful to be a disease diagnostic index, especially in cancer screening. As core subunit of cohesin, RAD21 up-regulation facilitates cohesin to load on chromatin, resulting in constant TAD length and increased space volume[78]. Yujie Sun group attributed it to full ring extrusion within TAD. While in sight of this work, over-expression of RAD21 could increase the complexity of TAD hierarchy. That provides the technical possibility to study key factors of TAD hierarchy and manipulate its structural alteration. Meanwhile,

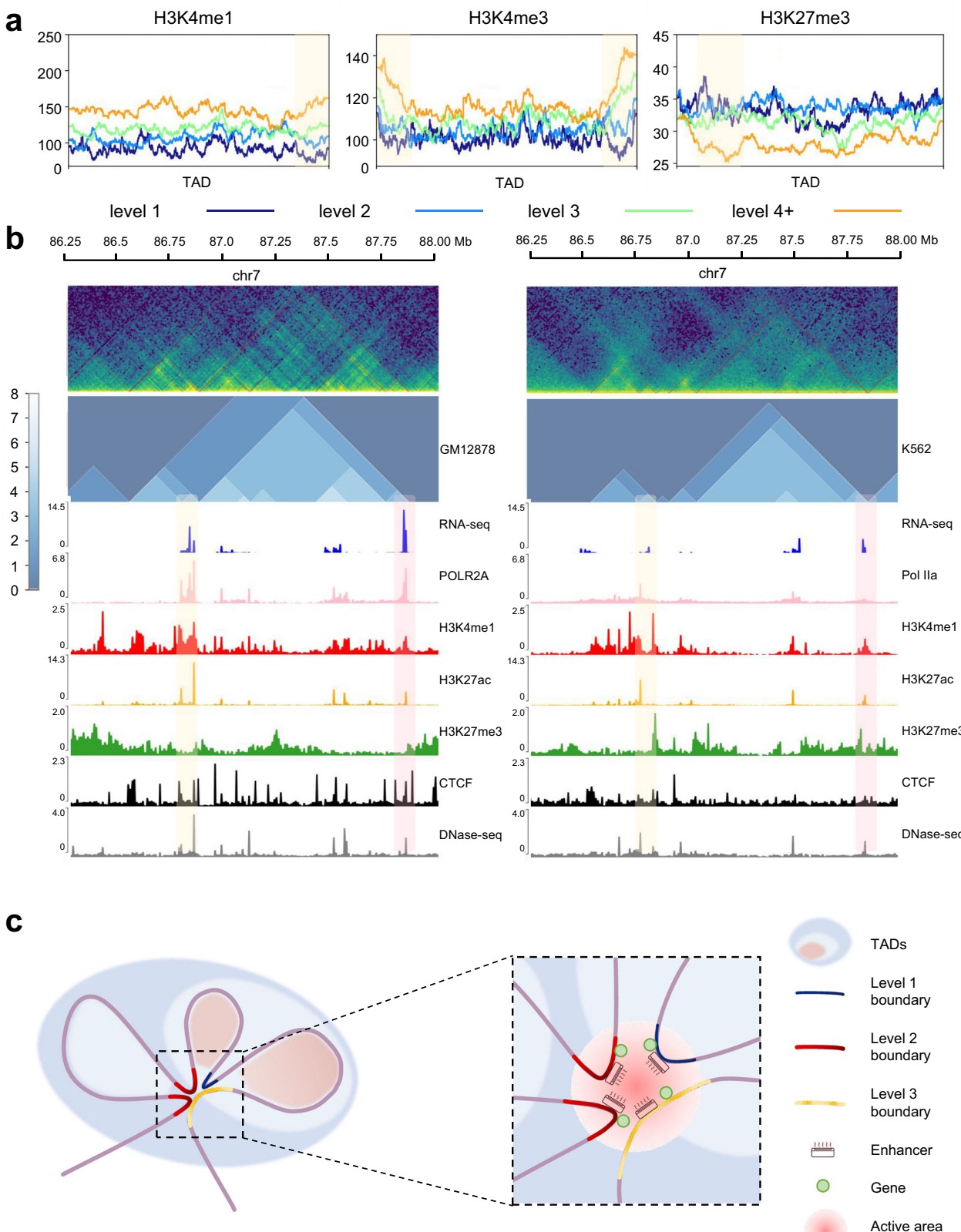

**Fig. 7 | Further analysis of TAD hierarchy with OnTAD. a** Enrichment of histone modifications within TADs and around boundaries of all levels. Level 1 (dark blue), level 2 (blue), level 3 (green) and level 4 plus (orange) are overlaid. The yellow shaded area indicates where the TAD boundary is located. **b** Representative example of TAD hierarchy and multi-omics landscape in GM12878 cell (left panel) and K562 cell (right panel). The upper green images show Hi-C heatmaps in both sides, and the middle blue images show the distribution of TAD hierarchy. Areas with yellow and pink shading are selected to depict inter-cellular variation. **c** Schematic representation of the air conditioner model for TAD hierarchy. TAD boundaries are marked according to the level.

increasingly more evidence have discovered that transcription factors and co-factors form phase-separated droplets at super-enhancers and exhibit powerful impact on gene control[79]. And phase separation of architecture protein Oct4 regulates TAD reorganization[80]. As component of TAD hierarchy, subTAD shows close connection with compartments and phase separation[27,37,81], partly contributing to the cell-to-cell variation. Similar with liquid-like condensates, high-level boundaries exhibit aggregation of functional elements and several genome zones. Thus, TAD hierarchy may hold the post of potential indicator of phase-separated condensates on Hi-C heatmaps.

On this account, future works of TAD hierarchy could focus on these aspects: (1) distribution preference of gene or regulatory elements in TAD hierarchy; (2) key molecules and their function on formation, maintenance, and reconfiguration of TAD hierarchy; (3) alteration of TAD hierarchy during embryonic development, phylogeny, viral infection, disease, tumorigenesis, and treatment; (4) reconfiguration of TAD hierarchy through DNA damage, repairing, and cell apoptosis; (5) relationship between TAD hierarchy and phase separation; (6) analysis of TAD hierarchy in single-cell data; (7) algorithm improvement combined with artificial intelligence and integrated analysis with multi-omics data.

Three-dimensional genome has achieved enormous progress these decades[82,83], of which the main approaches of capturing real chromatin structures are based on next-generation sequencing and micro-fluorescence imaging[84,85]. However, limiting to technical issues such as data type[86,87], sequencing depth, and resolution[88,89], there are still unknown regions to explore for the fine structure of 3D-genome and hierarchical TAD. The biggest dilemma in TAD hierarchy is that most studies and methods are based on bulk Hi-C data and next-generation sequencing[85], so there are still plenty of unresolved chromatin segments in genome 3D structures such as centromeres and telomeres, leading to the absence of solid standards of evaluation. Meanwhile, structural variations, such as insertion, deletion, copy number variations, translocation, can cause abnormal Hi-C heatmap and inaccurate TAD hierarchy recognition. Moreover, relevant researches in biology and medicine are numbered, further limiting exploration in TAD hierarchy. Hence, algorithm teams and experimental groups could utilize respective superiority and academic strengths. There is an urgency for more TAD hierarchy callers by improvement of algorithms, especially artificial intelligence (AI) including traditional machine learning and deep learning with artificial neural networks. AI is superior at learning patterns in massive amounts of published data to discover unknown information and improve the accuracy of predictions, rather than developing an algorithm to solve one problem. On the one hand, multiple machine learning methods have been well applied to studying DNA, RNA, proteins, and their interactions in the field of biology[90]. On the other hand, they are developed to effectively identify digenic traits and genotypes of specific diseases in genomics[91]. Another dominance is to make up for weaknesses of data, such as low sequencing depth, low resolution, data type, and background noise. For example, deep learning with interpretability trains a relatively superior model through learning complex relationships among various data types and could discover unknown information in the 3D-genome by enhanced resolution and structure recognition[88,89], which could supplement the blank of structure detection from next-generation sequencing or microscopic imaging. Particularly, that's beneficial to explore TAD hierarchical structure in single-cell data. Combination of AI and more excellent single-cell omics derivative techniques is supposed to resolve this problem and reveal relationship between chromatin spatial structure and transcriptional regulation in single cells, especially single-cell Hi-C (scHi-C)[92,93]. Besides, many machine learning models have been applied to detect spatial structures at different scales of genome[94,95], with promising usage in predicting TAD hierarchy. Furthermore, researchers of bioinformatics and genomics could make efforts to combine genome spatial architectures with better-fitting mathematical-physical models for higher sensitivity in interaction identification.

As for experiment groups, they could search for improvement in sequencing techniques, types of digestive enzymes, and experimental conditions for optimized preservation of chromatin structures. Therefore, technologies such as third-generation sequencing has more application prospects and will reveal more sufficient structural-regulatory characteristics of currently unresolved chromatin segments under the integration of the aforementioned AI technologies and multi-omics. Prospectively, more communication and cooperation between labs of multidisciplinary and deeper understanding of TADs will arouse interests of biomedical teams, and expand researches on TAD hierarchy in broader aspects of life activities including disease occurrence, development, and drug resistance.

## Methods
### Benchmark metrics
Similarities among callers, resolutions, sequencing depths, and matrices normalization are weighted by two metrics: Hier_SSIM and Overlap ratio.

1. To compare the similarity of TAD level recognition results, we defined the Hier_SSIM. Firstly, the results of the TAD callers were mapped to the matrix as large as Hi-C heatmap. The value in each pixel represents the TAD level of the pixel multiply by twenty. We used a sliding window with a size of 8 Mb to slide along the diagonal to intercept the TAD level map. Structural similarity (SSIM)[96] was used to evaluate the similarity of each map by skiamge Python package, and the average result was defined as Hier_SSIM.
2. Overlap ratio is calculated by SuperTAD version 1.2[40], by assessing the intersection of two TAD coding trees. Before that, we transform all hierarchical TAD results into form of SuperTAD outputs file.

As for similarity among resolutions for one method, we calculate mean of values.

### Preprocessing of datasets
Within this study, the majority of in situ Hi-C contacts are downloaded, filtered by $MAPQ \geq 30$, and extracted for intra-chromosomal contact (contacts within the same chromosome) matrices on chromosome 7. As for S2 cell line, we downloaded the paired SRA files from the Sequence Read Archive (SRA) under the accession number SRR9019613 by SRA Toolkit (http://www.ncbi.nlm.nih.gov/Traces/sra/, v3.0.10) with the prefetch tool (https://github.com/ncbi/sra-tools/tree/master/tools/prefetch), and converted SRA files to fastq files by the fastq-dump tool (https://github.com/ncbi/sra-tools/tree/master/tools/fastq-dump) on double-ended sequencing data. Then we used Hi-C-Pro[97] (v2.11.4) to obtain Hi-C contact matrix.

Hi-C contact matrix normalization was performed using the iterative correction and eigenvector decomposition method (ICE)[56] for inputs of most TAD hierarchy callers. Knight-Ruiz normalization (KR)[98] is conducted for inputs of Arrowhead and HiTAD. For each sample, we retained a copy of raw data. All samples were generated at five various resolutions: 5 Kb, 10 Kb, 25 Kb, 50 Kb, and 100 Kb, for the comparison of all TAD hierarchy callers. The resolution is equal to single bin sizes. When making multi-omics track graphs, we fix the bin size at 10 Kb, and make the sequencing depth of GM12878 the same as that of K562.

The bigWig format of ChIP-seq data and RNA-seq data of GM12878 and K562 were downloaded from the Encyclopedia of DNA Elements Encyclopedia of DNA Elements (ENCODE) project[99] (https://www.encodeproject.org/). We then calculate scores per genome regions with the computeMatrix command of deepTools[100] (parameter: scale-regions, regionBodyLength: 30000, skipZeros). The computed files of calculating were further used to generate profile with the plotProfile

command of deepTools. RNA-seq data of colorectal and paracancerous tissues are from J. Aryee and E. Bernstein (GEO accession number GSE133928[101]). We downloaded the TPM count files of BRD3187 and BRD3187N, made it to the expression matrix, and conducted vst normalization by DESeq2[102]. Then we matched it with the reference genome to fetch the gene coordinate and got the bedgraph for each sample. We finally obtained the bigWig file, by merging the overlap region with bedTools, counting the sum of expression of the gene involved as that of the overlap regions, and inputting bedgraph file to bedGraphToBigwig[103]. Specific accession numbers of these datasets are recorded in Supplementary Data 2.

The inputs of all callers were generated as separate requirements. The sparse file was a three-column files, containing the two bins that have interaction and contacts, similar to Rao format with coordinate replacing bin number in 1st and 2nd column. The dense file and the hic format were generated from the sparse file by 'sparseToDense' and 'hicpro2juicebox' of Hi-C-Pro. The cool files for comparison of sequencing depths were generated by cooler[104] from the sparse file. KR cool files as inputs of HiTAD were downloaded from published Hi-C datasets (ftp://cooler.csail.mit.edu/coolers). The cool files of GM12878, GM12878 replicate, CH12-LX, S2, and GM12878 single-cell Hi-C data were translated from the hic files (*.hic) by HiCExplorer[105]. And the catch file (*.dat) is just for CaTCH with 4 columns, equaling to adding number of chromosomes before the 1st column of the sparse file.

The single-cell Hi-C data of GM12878 and IMR90 obtain from Kim et al.[106]. were preprocessed according to the protocol described in Ramani et al.[107]. Briefly, raw fastq, inner barcode, and outer barcode were first download from the 4D Nucleome (4DN) project with accession number 4DNESUE2NSGS and 4DNES4D5MWEZ. Then, the adapter of raw fastq files was trimmed using SeqPrep (v1.2) software with parameter: -A AGATCGGAAGAGCGATCGG -B AGATCGGAA-GAGCGTCGTG. Next, clean reads were aligned to hg19-mm10 combo-reference using bowtie2[108] (v2.3.5.1). Output bam files were sorted using samtools[109] (v1.7) and converted to bed format using bedtools[110] (v2.26.0). To filtered low-quality data, all cellular indices with fewer than 1000 unique reads were removed and we filtered out all indices where the cis:trans chromosome interaction ratio was lower than 1. At last, bin_schic.py script was used to generate single-cell Hi-C contact matrix at 10 kb, 25 kb, 50 kb, and 100 kb resolutions.

To mix data, the raw contact map of Hi-C of different cell lines were downsampled to the same contact number according to the original contact distribution. We further downsampled the contact map as required and added the number of interactions at the same location on the contact map for different mixing ratios, including 100% GM12878, 100% K562, together with 9:1, 8:2, 7:3, 6:4, 5:5, 4:6, 3:7, 2:8 and 1:9 for GM12878 to K562. As for the ratio of GM12878, K562, and IMR90, the percentage setting is 1:1:1, apart from pure samples. Finally, we performed ICE correction on all mixed raw data for further analysis.

### Analysis for benchmarking

**Hierarchical level analysis.** All original outputs from various callers are uniformly transformed into three-column files, containing start bin/coordinate, end bin/coordinate, and TAD level. Then we calculate the max number of each TAD boundary in start and end column as its boundary level. Next, we filter TAD segments of which length is shorter than 30 Kb and larger than 2 Mb and compute the distribution of TAD and boundaries at all levels. Length comparison and genomic coverage are generated from files without filtration.

**Genomic coverage analysis.** Firstly, the complement segments of each hierarchical TAD output is obtained by BEDOPS[111]. Then, we calculate the sum length of these segments, and the ratio of it to the whole size of chromosome 7 from seven cell types (GM12878, HMEC, HUVEC, IMR90, K562, KBM7, and NHEK). And the final result is 1 minus their average. Specific results are recorded in Supplementary Data 1.

**Visualization.** The Hi-C heatmap, TAD hierarchy diagram, and multiple epigenomic tracks are pictured by HiCExplorer[105].

### Statistics & reproducibility

No statistical method was used to predetermine the sample size. No data were excluded from the analyses. The experiments were not randomized. The Investigators were not blinded to allocation during experiments and outcome assessment.

### Construction of a summary table

We constructed a summary table to show the performances of all callers (Fig. 5), including biological correlation, robustness, and usability. Biological correlation and robustness correctly show the main evaluation work as described above. As for usability, we concern difficulty of installation, complexity of input and output files, fluency of running code, computational memory and resource consumed, whether with built-in visualization, and whether with resolution self-identified. The degree of performance was presented by normal, good and excellent, shown in the image as grey dot, colorful dot, and colorful square. Since usability varies largely among methods, it covers all degrees, while there only includes two assessments in biological correlation and robustness.

### Implementation of methods

Arrowhead[7]: we used the code of Arrowhead v1.0.0 based on Juicer_tools v2.09.00[112] from https://github.com/aidenlab/juicer/wiki/Arrowhead. We set $m$ to 2000, which are the default parameter settings, and resolution parameter were set by data resolution.

Armatus[23]: we used the code of Armatus v2.3 from http://www.cs.cmu.edu/~ckingsf/software/armatus/. We set $\gamma$ max to 1 and $s$ to 0.05, and resolution parameter were set by data resolution.

CaTCH[42]: we used the code of CaTCH v1.0 from https://github.com/zhanyinx/CaTCH_R. We set $RI$ of subTAD to 55%, and that of TAD to 69%.

HiTAD[25]: we used the code of HiTAD v0.4.2 from https://pypi.python.org/pypi/TADLib. We set maxsize to 2000000, and resolution parameter were set by data resolution.

matryoshka[43]: we used the code of matryoshka v1.0 from https://github.com/COMBINE-lab/matryoshka. We set $\gamma$ max to 1 and $s$ to 0.05, and resolution parameter were set by data resolution.

OnTAD[38]: we used the code of OnTAD v1.2 from https://github.com/anlin00007/OnTAD.git. We set penalty to 0.1 as default, maxsz and minsz according to resolution (size range of TAD segment is 30 Kb-2 Mb).

Multi-CD[37]: we used the code of Multi-CD v0.1.0 from https://github.com/multi-cd/multi-cd-matlab.

IC-Finder[46]: we used the code of IC-Finder from http://membres-timc.imag.fr/Daniel.Jost/DJ-TIMC/Software.html.

TADpole[47]: we used the code of TADpole v0.0.09000 from https://github.com/3DGenomes/TADpole. We set resolution parameter were set by data resolution.

BHi-Cect[44]: we used the code of BHi-Cect v1.0.0 from https://github.com/princeps091-binf/BHi-Cect.

SpectralTAD[45]: we used the code of SpectralTAD v1.2.0 from https://bioconductor.org/packages/SpectralTAD/. We set qual_filter to FALSE as default, and levels to maximum effective value of each sample.

HBM[49]: we used the code of HBM v1.0.0 from https://github.com/yolish/hbm/.

spectral[50]: we used the code of spectral v1.0.0 from https://drive.google.com/file/d/1lfbtDyuVibwJkD-0n_Xy8fpjRmj25JrO/view?usp=sharing.

3DNetMod[48]: we used the code of 3DNetMod v1.0 from https://bitbucket.org/creminslab/3dnetmod_method_v1.0_10_06_17. The code were set as author commended.

GRiNCH[41]: we used the code of GRiNCH v1.0.0 from https://roy-lab.github.io/grinch/. We set expected_size_of_cluster to 500000, 1000000, and 2000000 respectively for subTAD, TAD, and metaTAD, which were commended by the author.

deDoc[51]: we used the code of deDoc v1.0.0 based on Java v1.8.0_92 from https://github.com/yinxc/structural-information-minimisation. No parameter was required to set.

SuperTAD[40]: we used the code of SuperTAD v1.2 from https://github.com/deepomicslab/SuperTAD. We chose deepbinary mode, and no parameter was required to set.

TADtree[33]: we used the code of TADtree v1.0.0 from http://compbio.cs.brown.edu/projects/tadtree/. We set $p = 3$, $q = 12$, and $gamma = 500$ as default. $S$ and $M$ were set according to data resolution (size range of TAD segment is 30 Kb-2 Mb).

GMAP[26]: we used the code of GMAP v1.4 from http://tanlab4generegulation.org/rGMAP_1.1.tar.gz. We set the following parameters: resl = $10 * 10^3$, logt = T, dom_order = 2, min_d = 25, min_dp = 5, hthr = 0.9, bthr = Bg_d, t1thr = 0.75, fcthr = 0.9. While resl, Max_d, Max_dp, and Bg_d were set according to data resolution.

PSYCHIC[54]: we used the code of PSYCHIC v1.0.0 from https://github.com/dhkron/PSYCHIC.

HiCKey[52]: we used the code of HiCKey v1.0 from https://github.com/YingruWuGit/HiCKey. We use default parameters.

## Computational resource

A total of 4 workstations and 5 server nodes are used for data processing.

Workstation 1: The workstation used to test all methods had an Intel(R) Xeon(R) CPU W-2175 @ 2.50 GHz (19,712 KB cache size; 28 cores in total) and 256 GB of memory. The GPUs were one Nvidia Quadro RTX5000 16 G. The operating system used was Ubuntu 20.04.2 LTS.

Workstation 2: The workstation used to test all methods had an Intel(R) Xeon(R) CPU W-2225 @ 4.10 GHz (8448 KB cache size; 8 cores in total) and 256 GB of memory. The GPUs were one Radeon RX550/550X. The operating system used was Ubuntu 18.04.6 LTS.

Workstation 3: The workstation used to test all methods had an Intel(R) Xeon(R) Gold 5218 R CPU @ 2.10 GHz (28,160 KB cache size; 40 cores in total) and 256 GB of memory. The GPUs were one NVIDIA RTX A6000 48 GB. The operating system used was Ubuntu 18.04.6 LTS.

Workstation 4: The workstation used to test all methods had an Intel(R) Xeon(R) Gold 6258 R CPU @ 2.70 GHz (39,424 KB cache size; 112 cores in total) and 256 GB of memory. The GPUs were one Llvmpipe 2684. The operating system used was Ubuntu 18.04.6 LTS.

Server nodes 1: The server node used to test all methods had an Intel(R) Xeon(R) Gold 5320 CPU W-2175 @ 2.20 GHz (39,936 KB cache size; 52 cores in total) and 256 GB of memory. The GPUs were one Nvidia Tesla A100 40 GB. The operating system used was Red Het 4.8.5–44.

Server nodes 2, 3, 4, and 5 have the same specifications: The server node used to test all methods had an Intel(R) Xeon(R) Gold 5320 CPU W-2175 @ 2.20 GHz (39,936 KB cache size; 52 cores in total) and 256 GB of memory. The GPUs were two Nvidia Tesla A100 40 GB. The operating system used was Red Het 4.8.5–44.

## Reporting summary

Further information on research design is available in the Nature Portfolio Reporting Summary linked to this article.

## Data availability

All relevant data supporting the key findings of this study are available within the article and its Supplementary Information files. A summary of the data used in this study is shown in Supplementary Data 2. In situ Hi-C data of GM12878, GM12878 replicate sample, HMEC, HUVEC, IMR90, K562, KBM7, NHEK, and CH12-LX cell lines are available in the GEO database under accession code GSE63525. Cool files of 5 cell types (HMEC, HUVEC, IMR90, K562, and KBM7) were downloaded from the Cooler [ftp://cooler.csail.mit.edu/coolers] before, but may be requested by email to the authors now. In situ Hi-C data of drosophila S2 cell line are available in the SRA database under accession code SRR9019613. ChIP-seq data and RNA-seq data of GM12878 and K562 are available in the ENCODE project[99] (https://www.encodeproject.org). CTCF ChIP-seq data of GM12878, K562, CH12-LX, and S2 cell are available in the ENCODE project under accession number ENCSR000DZN, ENCSR000DWE, ENCSR000ERM, and ENCSR711UTK. For GM12878 cell line, ChIP-seq data (H3K4me3, H3K27ac, POLR2A, H3K9ac, H3K27me3, H3K9me3, and H3K4me1) are available in the ENCODE project under accession number ENCSR057BWO, ENCSR000AKC, ENCSR000EAD, ENCSR000AKH, ENCSR000AKD, ENCSR000AOX, and ENCSR000AKF, respectively. RNA-seq data and Dnase-seq data are available in the ENCODE project under accession number ENCSR843RJV, and ENCSR000EMT, respectively. For K562 cell line, ChIP-seq data (H3K27ac, POLR2A, H3K27me3, and H3K4me1) are available in the ENCODE project under accession number ENCSR000AKP, ENCSR031TFS, ENCSR000EWB, and ENCSR000EWC, respectively. RNA-seq data and Dnase-seq data are available in the ENCODE project under accession number ENCSR594NJP, and ENCSR000EKS, respectively. For paired colorectal and paracancerous tissues, Hi-C data and RNA-seq data of are available in the GEO databse under accession code GSE133928, from which we downloaded the TPM count files of BRD3187 and BRD3187N. The single-cell Hi-C data of GM12878 and IMR90 are available in the 4DN project[113] (https://www.4dnucleome.org/) under accession code 4DNESUE2NSGS and 4DNES4D5MWEZ. Source data are provided with this paper.

## Code availability

The analysis code is available both on GitHub and Zenodo under the following link [https://github.com/XiangXuCode/TAD_hierarchy_benchmark] and [https://doi.org/10.5281/zenodo.10982207][114].

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

## Acknowledgements

X.X., X.Bo, and H.C. were supported by National Natural Science Foundation of China (62173338) and Beijing Nova Program of Science and Technology (20220484198). J.X., D.H., and J.G. were supported by National Natural Science Foundation of China (82073223). We thank Professor Cheng Li, Professor Hao Li, Yu Sun, Huan Tao, Yang Ding, Kang Xu, Longteng Wang, and Mengge Tian for insightful feedback and discussions.

## Author contributions

J.X. and X.X. performed the majority of analyses, including TAD hierarchy prediction and similarity comparison. H.C., J.G., and X.Bo conceived the project. H.C. and D.H. wrote the manuscript. Y.L. did a lot of computational work in revising the manuscript. L.L. compiled the results of each method and performed a structural variance analysis. Y.Z. and D.H. prepared schematic diagrams. Y.L., L.L., Q.Y., and Y.C. downloaded and installed all callers. X.Bai, A.H., J.S. searched all the tools and literature. Y.L., X.Bai, and L.L. proofread the manuscripts. All authors read and approved the final manuscript.

## Competing interests

The authors declare no competing interests.
