## [Peer Review File · Nature Communications]

A comprehensive benchmarking with interpretation and operational guidance for the hierarchy of topologically associating domainsReviewer #1 (Remarks to the Author):

General:

In their work, Xu et al. conducted a comparative analysis of 13 computational tools to address the issue of result inconsistency among various algorithms. They compared outputs across different data resolutions, sequencing depths, and matrix normalizations. Additionally, they proposed a model to emphasize the role of TAD hierarchy in transcription while evaluating the hierarchy similarities between different TAD detection algorithms.

Here are my comments for the authors:

1. The authors mention a motivation for exploring the existence of TAD hierarchy in single-cell data. However, before addressing this, it's crucial to determine if these tools are suitable for finding TADs in single-cell Hi-C data, given that most of them were tested on bulk Hi-C data. Despite appearing technically feasible, the sparsity of single-cell data might pose challenges. The authors should consider including an analysis that demonstrates the suitability of these tools for such data.
2. Figure 1a does not provide an exhaustive list of TAD algorithms to date. While paragraph 1 of the "Compendium of hierarchical TAD callers" section may imply that it does, algorithms such as EAST, ClusterTAD, CHDF, or IS were not included. Additionally, Figure 1a could benefit from clearer labeling and a more user-friendly design. The authors added a simple illustration to explain the groupings, for instance there is a linear regression curve below the statistical model figure, I think the author should have a thin line pointing from the name to the illustration for all the different classes. Also, there was no illustration provided for the structural entropy class out of the 5 groups provided.
3. It is unclear what Figure 1b is trying to convey. Proper labeling of TADs, sub-TADs, and a clear purpose for the figure would enhance its comprehensibility.
4. In Line 146, the authors mentioned their intention to use Hi-C dataset GSE63525. However, GSE63525 is an accession number unique to the GEO repository. It should be clarified that they intend to use the Hi-C data available under the Gene Expression Omnibus (GEO) accession number GSE63525.
5. The "Compendium of hierarchical TAD callers" and "Evaluation among callers and within each caller" sections appear to lack relevant content, including analysis figures and citations that correspond to their suggested titles. This should be addressed.
6. In paragraph 1 of the "Variation of hierarchical TAD structures among various callers," it's unclear whether the authors refer to two hypothetical TADs from different algorithms or the same algorithms. This point was not adequately explained in the article or the accompanying figure.
7. Regarding the section titled 'Variation of hierarchical TAD structures among various callers':
 - a. Can the authors elaborate on the criteria and methodology used for grouping the algorithms? While hierarchical clustering was mentioned, an explanation of the observed variations in enrichment results within the same group would help clarify the consistency of this grouping.
 - b. How do the variations in enrichment results among algorithms within the same group impact the overall interpretation of the hierarchical TAD structure analysis? Do they suggest a need for refining the grouping criteria or considering other factors influencing algorithm performance?
8. The box plots in Figure 3a and 3b are too small to read even at 200% zoom.
9. In Figure 5a, it is unclear what the squares or circles represent.
10. Recently, there have been several papers that explore comparative studies of TAD algorithms. While this article introduces the possibility of exploring single-cell Hi-C data, it lacks a comprehensive comparative analysis to establish the unique advantages and distinctions of using this data. The statement in the paper, "The choice of Hi-C data or single-cell Hi-C data may influence outputs and performance to a large extent, but this does not represent the absolute value of these tools," leaves some uncertainty regarding the potential benefits, which the article should have addressed. In an analysis article, readers often seek a clear and objective assessment of the methods or approaches presented. A more detailed comparative analysis could provide valuable insights for better understanding the implications and utility of adopting this approach.

Minor:

1. In Table 1, the authors should provide notes explaining the definition of a sparse matrix to clarify it for readers. It appears that they refer to an N by N symmetric matrix, but this is not explicitly stated in the article. The same goes for the use of a sparse format.
2. There are a few grammatical errors that should be corrected, such as:
 - a. Line 76- 78: Grammar needs to be fixed here. " Recent researches collectively refer to TAD, subTAD and metaTAD as TAD hierarchy, of which the level is characterized by positively correlated with CTCF enrichment, gene activity, gene density, and active epigenetic states" Could be: Recent research collectively refers to TAD, subTAD, and metaTAD as TAD hierarchy, with the level characterized by positive correlations with CTCF enrichment, gene activity, gene density, and active epigenetic states.
 - b. Line 100: Grammar needs to be fixed. Increasingly more "researches" structure to date proble into sub-TAD.. The use of researches is not correct.
 - c. Line 255: In summary..

Reviewer #2 (Remarks to the Author):

Xu et al performed a benchmarking analysis of Topologically Associating Domain (TAD) callers using in situ Hi-C data (by Rao et al., 2014). Despite the commendable effort and detailed inclusion of references and graphical data, the manuscript markedly falls short in establishing its novelty and adhering to the rigorous scholarly writing caliber requisite for publication in Nature Communications.

The manuscript gravely lacks clarity and robustness in illustrating its novelty and distinctiveness from prior benchmarking studies, including those by Zufferey et al. (2018) and Liu et al. (2012). TAD's nature as a model without a well-defined ground truth makes benchmarking the algorithms challenging. And comparing benchmarking methods is even more so. The paper does not sufficiently navigate or address these complexities and thus, fails to carve out a meaningful new niche in the existing literature.

Additionally, there are major mistakes in the design of analyses such as conflation of "house-keeping genes" with protein-coding genes and utilizing bulk-level, cell type-specific features to rationalize single-cell phenomena.

Finally, The analogy relating to air conditioners is not intuitive or helpful. Numerous typos and lexical errors exist, for instance:

- Line 50: "conservative" → "conserved"
- Line 59: "Dynamic" → "Dynamics"
- Line 81: Perhaps, "as a possible mechanism"
- Line 117: A grammatical adjustment is necessary, potentially "features assume."
- Line 423: Adjustment may be "expression. Perhaps"

Reviewer #3 (Remarks to the Author):

In this paper, the authors performed a comprehensive benchmark study of the 13 algorithms to define TAD hierarchy. This work addressed an important research question. The results are solid and convincing. The authors provided some useful guidelines for practitioners how to select the optimal computational methods. The paper is also well written. I feel this work has potential to address broad interest from readers of Nature Communications, and researchers in 3D genome field. I just have a few comments and suggestions which may further improve the current manuscript. Here are my specific comments.

Major comments:

1. The authors applied 13 methods to Hi-C data from 7 human cell lines (GSE63525), and only tested chr7. For the completeness of results, I hope the authors can perform genome-wide analysis to identify TAD hierarchy in all chromosomes. In addition to human data, I hope the authors can apply the 13 methods to mouse Hi-C data and drosophila Hi-C data, and evaluate whether their conclusions from the benchmark study can also be applied to mouse and drosophila.
2. Page 6, line 147, the authors claimed to benchmark 13 methods. However, Fig 2b, c, d only contain 11 methods. TADtree and TADpole are missing. The authors need to add the results of TADtree TADpole in Fig 2 and other main figure / supplementary figures.
3. In some figures (e.g. Fig 2c, Fig 7b, Fig S10, etc), I suggest the authors to present both Hi-C contact heatmap and inferred TAD hierarchy, to facilitate easy visual inspection and interpretation.
4. Fig 3, why some methods have less than 5 bin resolutions?
5. Methods section: in the definition of Hier_SSIM, the authors used a sliding window of 4Mb. I hope the authors can justify the selection of 4Mb. Whether the conclusions are robust for different size of sliding windows? Some sensitivity analysis would be very informative.
6. In the analysis related to Fig S8, the authors simulated the cellular heterogeneity by mixing GM12878 and K562. A better way is to create pseudo bulk Hi-C data from scHi-C data from GM12878 and IMR90, etc (PMID: 32946435), with different mixture proportions.
7. What is the reproducibility of TAD hierarchy calling results? If applying each of 13 methods to biological replicates of Hi-C data from the same cell type, which methods can achieve the most reproducible results?
8. Fig 7b, since the authors compare normal cell line GM12878 with cancer cell line K562, whether the difference is due to structural variations, such as insertion, deletion, copy number variations, translocation, in K562 cells?

Minor comments:

1. The authors need to provide more details of running time and memory cost for each of 13 methods.
2. It seems that Fig 7c is missing?
3. Typo. Page 8, line 210: "with 2M" should be "within 2Mb". Line 211: "within 1M" should be "within 1Mb".
4. Typo. Legend in Fig 5d "Raio" should be "Ratio".
5. The authors considered 5 bin resolutions (5Kb, 10Kb, 25Kb, 50Kb, 100Kb). Page 11, line 290: matryoshka also performs well on low-resolution data (~500Kb). Is "500Kb" a typo of "50Kb"?

Revision

Reviewer #1:

General:

In their work, Xu et al. conducted a comparative analysis of 13 computational tools to address the issue of result inconsistency among various algorithms. They compared outputs across different data resolutions, sequencing depths, and matrix normalizations. Additionally, they proposed a model to emphasize the role of TAD hierarchy in transcription while evaluating the hierarchy similarities between different TAD detection algorithms.

RESPONSE:

Thank you for your meticulous review and affirmation of our research. And we have made the following improvements to this manuscript based on your great and constructive suggestions.

Major comments:

1. The authors mention a motivation for exploring the existence of TAD hierarchy in single-cell data. However, before addressing this, it's crucial to determine if these tools are suitable for finding TADs in single-cell Hi-C data, given that most of them were tested on bulk Hi-C data. Despite appearing technically feasible, the sparsity of single-cell data might pose challenges. The authors should consider including an analysis that demonstrates the suitability of these tools for such data.

RESPONSE:

We are very grateful for your valuable suggestions on our research of TAD hierarchical structure in single cells.

Indeed, the sparsity of single-cell data might pose challenges in TAD hierarchical structure recognition. Therefore, we performed these TAD hierarchy recognition algorithms on GM12878 single-cell Hi-C data (Tan et al. Science 2023).

As a result, we noticed a low correlation between the TAD hierarchical structure recognized by single-cell Hi-C and the recognition results of bulk Hi-C (Figure 1 for reviewer), which is consistent with your concerns. We noticed that the correlation between single-cell and bulk identification results is poor, and the only method that can cluster together, GRiNCH, has a low correlation between bulk and single cell. This indicates that existing methods have poor performance in identifying TAD hierarchy at the single-cell level. This may be due to the sparseness of single-cell Hi-C data.

Figure 1 for reviewer

On the other hand, to better summarize our research content, we revised the title in Part 7 from “TAD hierarchy tends to be structure at single-cell level” to “The impact of cell heterogeneity on TAD hierarchy”. In this part, we tended to explore the formation mechanism of TAD hierarchical structure. In our previous research work, we found that there are two possibilities for the formation of TAD hierarchical structures (Figure 2 for reviewer, PMID:33897976).

Figure 2 for reviewer, PMID:33897976

In the first hypothesis of Figure 2A, the TAD hierarchy is absent in a single cell. the TAD hierarchy arises as a result of mixing between different cells, resulting in a heterogeneous TAD superposition between the different cells. According to this hypothesis, then the result of mixing two different cellular Hi-Cs would, with high probability, lead to the complication of the TAD hierarchical structure, resulting in a statistically elevated TAD hierarchy. On the contrary, if the TAD hierarchy structure is not due to heterogeneous TAD superposition between cells, then the mixing should result in the TAD hierarchy being in between the two cells.

Our results in this manuscript show that the second hypothesis in Figure B is more likely to be the mechanism of TAD hierarchy formation. The above is what we would like to elaborate in this part. We are very sorry that our presentation has caused you to misunderstand.

Thank you very much for your careful review of our work, which makes our conclusions clearer and more reliable.

2. Figure 1a does not provide an exhaustive list of TAD algorithms to date. While paragraph 1 of the "Compendium of hierarchical TAD callers" section may imply that it does, algorithms such as EAST, ClusterTAD, CHDF, or IS were not included. Additionally, Figure 1a could benefit from clearer labeling and a more user-friendly design. The authors added a simple illustration to

explain the groupings, for instance there is a linear regression curve below the statistical model figure, I think the author should have a thin line pointing from the name to the illustration for all the different classes. Also, there was no illustration provided for the structural entropy class out of the 5 groups provided.

RESPONSE:

We are very grateful for your careful investigation of the research background of our work.

In our manuscript, we classified the existing TAD recognition methods into two categories, one is the algorithms that can only obtain single-layer TAD structures, and the other is the algorithms that can obtain multi-layer TAD structures. Methods such as EAST, ClusterTAD, CHDF, or IS that you proposed belong to the single-level TAD structure recognition algorithms. Since single-level TAD ignores the differences in interactions between TADs, it is inadequate for genomic structure feature extraction. With the proposal of TAD multilayers, such algorithms take into account the regulatory relationships that exist between subTADs and find a correlation between TAD layers and epigenetic inheritance and expression, thus TAD layer structure is biologically important.

There has been a lot of work evaluating TAD recognition algorithms (Zufferey et al. (2018) and Liu et al. (2012)), but few algorithms for recognizing more biologically significant TAD hierarchies, so this paper focuses on the TAD hierarchy recognition algorithms and how well they recognize the hierarchies. Algorithms for single-layer TAD structures such as EAST, ClusterTAD, CHDF, or IS are not included in the methods we will evaluate due to their lack of ability to detect TAD hierarchies.

Next, we greatly appreciate your advice on the understandability and aesthetics of our main Figure 1a.

Indeed, as you mentioned, we draw the schematic in Fig. 1a to let readers better understand the underlying principle of our algorithm for dividing the TAD hierarchy recognition. We are sorry for missing illustration and schematic diagram of structural entropy.

Therefore, we added an illustration of structural information entropy as follows. "The structural entropy is defined over the coding tree of a graph by fixing and decoding the graph in a way that minimizes the uncertainty occurring in random walks in the graph. This means that the structural entropy of the graph is the information embedded in the graph that determines and decodes the essential structure of the graph."

And, we draw the following schematic diagram based on the principle of structural information entropy for representing the principle of the algorithm of structural information entropy.

After we optimized the Figure 1a, the results are shown below:

Figure 1a

We sincerely thank you for your valuable comments on the results of Fig. 1, which deepened our insights into the information entropy algorithm and provided the reader with a more user-friendly schematic.

3. It is unclear what Figure 1b is trying to convey. Proper labeling of TADs, sub-TADs, and a clear purpose for the figure would enhance its comprehensibility.

RESPONSE:

Thank you for your suggestion, we apologize for the confusing layout of Figure 1b and the lack of clarity of the figure notes.

In Figure 1b, we want to illustrate how the TAD layer number and the TAD boundary layer number are defined and calculated. However, it is considered that readers from different research fields may not have an in-depth knowledge of the definition and scaling of TAD structures. Therefore, we tried to show the size and positional relationship between TAD hierarchical structures and chromosomes with a schematic diagram in Figure 1 to clarify the research objectives.

However, our typographical, figure note and manuscript writing problems made Figure 1b difficult to

understand. Therefore, we have redrawn and reformatted the main figure 1. We have also labeled the TAD and subTAD as you suggested.

And we revise the figure legend: "Definition and calculation of TAD hierarchy. Top: Structure of TAD at chromosome and A/B compartment scales. Middle: spatial morphology of TAD and subTAD in the nucleus, light purple represents TAD, dark purple represents subTAD, dark yellow represents TAD boundary, light yellow represents subTAD boundary. Lower: TAD hierarchical structure in Hi-C heatmap, the color is consistent with the middle figure. The darker purple color represents higher TAD layers. Darker yellow represents higher TAD boundary hierarchy."

Figure 1

Finally, we have modified the original text line 131 for Fig. 1b: "TAD is a chromatin structure at the submegabase scale, which is shown as an isosceles right triangle significantly above the background in Hi-C thermograms. the TAD hierarchical structure is shown as a nested isosceles right triangle (Fig. 1b). To assess how many layers are nested in the TAD structure, we define level 1 for the TADs that don't

belong to any larger TAD outside, and the level increases by 1 as smaller TADs position an inner layer. For boundaries, the rule follows that of OnTAD: the maximum of TADs it belonging to in a single direction, suggesting one boundary may belong to different numbers of TADs by left and right (Fig. 1b)."

Thanks for the contractive advice, which makes the purpose and presentation of Figure 1b much clearer.

4. In Line 146, the authors mentioned their intention to use Hi-C dataset GSE63525. However, GSE63525 is an accession number unique to the GEO repository. It should be clarified that they intend to use the Hi-C data available under the Gene Expression Omnibus (GEO) accession number GSE63525.

RESPONCE:

Thanks for your careful checking and terrific suggestions.

We did overlook this crucial detail and defaulted to the reader knowing that the GSE number is the data encoding in the GEO database. We fully agree with and adopt your suggestion to change line 146 to: "We intend to use the Hi-C data available under the Gene Expression Omnibus (GEO) accession number GSE63525".

Thanks for this detailed suggestion, and we have revised similar errors throughout the entire manuscript.

5. The "Compendium of hierarchical TAD callers" and "Evaluation among callers and within each caller" sections appear to lack relevant content, including analysis figures and citations that correspond to their suggested titles. This should be addressed.

RESPONCE:

Thank you for reminding the problems with the manuscript.

Since other reviewers have also mentioned problems with missing images in the manuscript, we first surmised that the manuscript you received may be incomplete or the software used to open it may be incompatible, which may have changed the formatting of some of the content. Therefore, we recommend that you use Google Chrome or Firefox to view the PDF file.

Secondly, considering the lack of other methodological reviews in question 2 and the unclear meaning of "Variation of hierarchical TAD structures among various callers" in question 6, we assume that you have suggestions to add to the content of our manuscript. We assume that you are suggesting additions to the content of our manuscript. If you are sure that the manuscript is fully displayed, we hope that the answers to questions 2 and 6 will clear up your doubts and concerns.

Thank you very much for your suggestions, and if the manuscript is indeed unavailable due to formatting, please do not hesitate to communicate with the editors and us, and we will try to submit the manuscript to you promptly by other means.

6. In paragraph 1 of the "Variation of hierarchical TAD structures among various callers," it's unclear whether the authors refer to two hypothetical TADs from different algorithms or the same algorithms. This point was not adequately explained in the article or the accompanying figure.

RESPONSE:

We would like to thank you for pointing out the presentation problem of our manuscript.

Indeed, our title "Variation of hierarchical TAD structures among various callers," tends to confuse the reader. Actually, we referred to "two hypothetical TADs from different algorithms".

In this part of the results, we would like to compare the differences in the recognition results of different algorithms under the same cell line GM12878 and the same resolution of 10Kb. Therefore, in this part, we use the same input data and calculate the outputs of different methods, if the TAD hierarchical structure of the outputs of the two methods are similar, then the higher the value of Hier_SSIM. In this way, we found that the four classes of algorithms, matryoshka, SpectralTAD, Arrowhead, and deDoc, have a high similarity in recognition results (Figure 2b).

Figure 2b

To eliminate this misunderstanding, we change the title of this section to "Variation of hierarchical

TAD structures from different algorithms".

Thank you again for your careful review of our manuscript, and the modification of the presentation of this section can reduce the ambiguity of the manuscript.

7. Regarding the section titled 'Variation of hierarchical TAD structures among various callers:

a. Can the authors elaborate on the criteria and methodology used for grouping the algorithms? While hierarchical clustering was mentioned, an explanation of the observed variations in enrichment results within the same group would help clarify the consistency of this grouping.

b. How do the variations in enrichment results among algorithms within the same group impact the overall interpretation of the hierarchical TAD structure analysis? Do they suggest a need for refining the grouping criteria or considering other factors influencing algorithm performance?

RESPONSE:

Thanks for your great suggestion.

a. Our classification of methods is mainly based on the logical construction or underlying idea of the algorithm developer. Just the convergence and difference of the development process, even the various methods under the same development logic, they use different specific means, for example, in the statistical distribution simulation method, the method of zero distribution and binomial distribution will definitely be different.

b. These are classifications based on development ideas, and there is no complete correlation between the difference in results and the performance of the algorithm. Moreover, the algorithm recommendation for users that we sorted out later only involves the selection of individual methods. Due to significant differences between methods, it is currently difficult to find interpretable differences from clustering. The current classification methods contain algorithmic essence, which can guide researchers to develop new algorithms from a theoretical perspective and improve recognition accuracy. Therefore, we retained the grouping criteria.

8. The box plots in Figure 3a and 3b are too small to read even at 200% zoom.

RESPONSE:

Thank you for your careful review and great suggestions.

Indeed, when we plotted Figure 3a, we only focused on the pattern of change with resolution and neglected to show the subtleties of the figure. In order to make our figure show the results more clearly, we made the following changes:

The vertical axis of Figure 3a was changed from TAD length to $\log_{10}(\text{TAD length})$, and we changed the figure to bitmap format (Figure 3a, Figure S5).

Figure 3a

Shown in Figure 3b is the distribution of the number and proportion of TAD hierarchical structures in different algorithms and in different cell lines. Since some algorithms identified few or no TADs at level 4+, it is difficult to tell from the figure. We made the following modifications: put the proportional distribution into the attached figure, enlarged the bar graph in the main figure and moved some figures to supplementary figure. The results obtained are as follows:

Figure. 3b, 3c

Number of TAD region at each level

Percent of TAD region at each level

Figure. S5

9. In Figure 5a, it is unclear what the squares or circles represent.

RESPONSE:

Thanks for your detailed suggestions. We apologize for the lack of clarity in the figure notes.

In Fig. 5a, it is our experience obtained by successfully reproducing the existing algorithms, and we hope to be able to use this part of the results as a reference for related researchers. We have three elements in Fig. 5a: grey circles, colored circles, and colored squares, which correspond to the three levels of normal, good, and excellent in the implementation.

We modify the figure note as follows: (Figure 5a)

Figure. 5a

Figure 5. Comprehensive evaluation of TAD hierarchy callers. (a) Summary performance of methods.

The grey circles, colored circles, and colored squares represent normal, good and excellent respectively.

10. Recently, there have been several papers that explore comparative studies of TAD algorithms. While this article introduces the possibility of exploring single-cell Hi-C data, it lacks a comprehensive comparative analysis to establish the unique advantages and distinctions of using this data. The statement in the paper, "The choice of Hi-C data or single-cell Hi-C data may influence outputs and performance to a large extent, but this does not represent the absolute value of these tools," leaves some uncertainty regarding the potential benefits, which the article should have addressed. In an analysis article, readers often seek a clear and objective assessment of the methods or approaches presented. A more detailed comparative analysis could provide valuable insights for better understanding the implications and utility of adopting this approach.

RESPONSE:

Thanks for your constructive suggestions.

With the development of single-cell sequencing technology, single-cell Hi-C allows us to capture chromatin 3D structures in individual cells, providing the possibility to probe the existence, prevalence, and dynamics of TAD hierarchical structures in individual cells.

However, due to the sparsity and high noise in single-cell Hi-C data, only DeDoc2 is currently able to identify TAD hierarchical structures in single cells (PMID: 37162225). In the DeDoc2 article, the authors evaluated the effectiveness of the bulk Hi-C-based TAD hierarchy algorithm for recognition in single-cell Hi-C using data from simulated single-cell sequencing model (Figure 3 for reviewer) .

Figure 3 for reviewer, PMID: 37162225

Therefore, with the current research progress, we recommend researchers to use DeDoc2 software when seeking single-cell TAD hierarchical structures. In the future, the development of single-cell Hi-C-based TAD hierarchy recognition algorithms may be able to further improve the accuracy of recognition.

Indeed, in the part of the text mentioning the possible advantages of single-cell Hi-C recognition of TAD hierarchies, we expressed it vaguely. Therefore, we have changed the description in our manuscript, "The input of single-cell Hi-C data provides a more realistic three-dimensional structure of chromatin, but at the same time poses the problem of high noise and data sparsity."

Many thanks for your wonderful suggestion, we recognized the importance of this suggestion for the evaluation of algorithms for TAD hierarchical structure, however, as there is currently only one algorithm, DeDoc2, that recognizes single-cell TAD hierarchical structure, researchers' choices are more limited. The emergence and evaluation of new algorithms for recognizing single-cell TAD hierarchies will be a trend in the future.

Minor comments:

1. In Table 1, the authors should provide notes explaining the definition of a sparse matrix to clarify it for readers. It appears that they refer to an N by N symmetric matrix, but this is not explicitly stated in the article. The same goes for the use of a sparse format.

RESPONCE:

Thanks for your great suggestion. We have provided notes explaining the definition and use of sparse matrix as followed:

“A matrix is a two-dimensional data object made of m rows and n columns, therefore having total $m \times n$ values. If most of the elements of the matrix have 0 value, then it is called a sparse matrix. In Hi-C data, sparse matrix represents chromatin contact map, the numerical values in row i and column j represent the frequency of DNA interaction between i bin and j bin in chromosomes. The sparse matrix is one of the common inputs for TAD hierarchical structure recognition algorithms.”

2. There are a few grammatical errors that should be corrected, such as:

a. Line 76- 78: Grammar needs to be fixed here. “Recent researches collectively refer to TAD, subTAD and metaTAD as TAD hierarchy, of which the level is characterized by positively correlated with CTCF enrichment, gene activity, gene density, and active epigenetic states” Could be: Recent research collectively refers to TAD, subTAD, and metaTAD as TAD hierarchy, with the level characterized by positive correlations with CTCF enrichment, gene activity, gene density, and active epigenetic states.

b.Line 100: Grammar needs to be fixed. Increasingly more “researches” structure to date proble into sub-TAD.. The use of researches is not correct.

c. Line 255: In summary..

RESPONCE:

Thanks for your careful review and detailed suggestions. We have revised all grammar vocabulary errors in our manuscript.

Thanks again for all your great and constructive suggestions.

Reviewer #2:

The manuscript gravely lacks clarity and robustness in illustrating its novelty and distinctiveness from prior benchmarking studies, including those by Zufferey et al. (2018) and Liu et al. (2012). TAD's nature as a model without a well-defined ground truth makes benchmarking the algorithms challenging. And comparing benchmarking methods is even more so. The paper does not sufficiently navigate or address these complexities and thus, fails to carve out a meaningful new niche in the existing literature.

RESPONCE:

Thanks for your constructive suggestions.

In fact, the research object, research question, research methodology and research results of this manuscript are different from the previous studies of Zufferey et al. (2018) and Liu et al. (2012).

First, our research object is the recognition algorithm of TAD hierarchical structure, while previous studies focus on the TAD recognition algorithm. the TAD hierarchical structure, relative to the boundary of the traditional TAD contains more feature information, and is more dynamic during the process of life development and differentiation. Therefore, we focus on the recognition algorithm of TAD hierarchical structure, which is more helpful to study the dynamic change law of chromatin in the cell nucleus under different conditions.

Next, since traditional algorithm evaluation focuses more on TAD structure boundaries, the scientific question that these studies want to address is whether the algorithms are robust to the localization of TAD boundaries. However, since our study focuses more on the TAD hierarchical structure, that is, under some conditions the TAD boundary position does not change but the TAD hierarchical structure changes. Therefore, our research question focuses on the effect of the recognition algorithm on the TAD hierarchical structure.

Motivated by a different research question, our research methodology was innovative compared to previous work. Traditional research and evaluation methods mainly utilize the conservativeness of TAD boundaries and TAD intervals. However, our study had to focus on nested TAD structure conservatism to evaluate the performance of these algorithms for recognizing aspects of TAD hierarchical structures. In previous studies, there is no evaluation of the performance of algorithms in terms of recognizing TAD hierarchical structures, therefore we quantify the TAD hierarchy and evaluate it with SSIM, which is a new evaluation method to assess the algorithms' recognition of TAD hierarchical structures.

Finally, in terms of the conclusions of the study, we not only give new conclusions about the performance of the algorithms in recognizing TAD hierarchical structures, but also our article aims to provide researchers with recommendations for choosing algorithms, and through our replication of all the tools we give the advantages and disadvantages of each algorithm and the difficulties of the implementation process.

Indeed, as stated by the reviewers, there is currently no gold standard for the identification and evaluation of TAD structures, this is that current chromatin conformation capture methods are indirect low-resolution single-moment methods of observing chromatin structure, which contain noise, bias, and batch effects, which pose challenges to the identification of TAD structures. In this case, we tested the effectiveness of TAD hierarchical structure identification methods by assessing the robustness and consistency of the identification using different methods on the one hand (commonly used in previous studies (Zufferey et al. (2018) and Liu et al. (2012))); on the other hand, it was an auxiliary validation using other multi-omics data, ATAC-seq, and CTCF ChIP-seq (Figure 2d) to ensure that the results of our

identification have a biological significance.

We apologize for not making the innovation and significance of our article clear to you earlier. We hope our further explanation can give you a new perspective on our manuscript

Additionally, there are major mistakes in the design of analyses such as conflation of "house-keeping genes" with protein-coding genes and utilizing bulk-level, cell type-specific features to rationalize single-cell phenomena.

RESPONSE:

Thanks for your careful review and critical suggestions.

First, we would like to apologize for conflation of "house-keeping genes" with protein-coding genes. Actually, we would like to study the enrichment of protein-coding genes in TAD boundaries, and we have revised the manuscript.

Next, we presume that you were questioning the questions and methods we wanted to study in the section of results 7. We apologize for the unclear description of the results in this section of the manuscript. Our research content in results 7 should be "The impact of cell heterogeneity on TAD hierarchy". We would like to explore the formation mechanism of TAD hierarchical structure. Here, we sincerely explain to you our original intention of doing this part of the results.

Although current advances in single-cell technology have made it possible to capture the spatial structure of chromatin within individual cells using single-cell Hi-C, however, because single-cell data is very sparse and contains a high level of noise, the multiple rectangles nested in bulk Hi-C heatmaps of TAD hierarchical-like structures are almost invisible in the raw data, and the data is usually processed by interpolation, etc., so that the identified TAD hierarchical structure authenticity is controversial. Therefore, there is no experimental means to accurately observe the presence or absence of TAD hierarchical structures in the nucleus.

However, it is important to explore whether TAD hierarchies are present in single cells, because if TAD hierarchies are not present in single cells, then it implies that TAD hierarchies are the result of superimposed cellular heterogeneity (Figure 4 for reviewer A). This will affect our understanding of the mechanisms of development and disease occurrence.

Figure 4 for reviewer, PMID:33897976

Therefore, we creatively came up with ways to simulate cell mixing to test both hypotheses A and B.

In the first hypothesis of Fig. A, the TAD hierarchical structure does not exist in a single cell. the TAD hierarchical structure arises due to the mixing between different cells, which results in the superposition of heterogeneous TADs between different cells. According to this hypothesis, then the result of mixing of two different cellular Hi-Cs would, with high probability, lead to the complication of the TAD hierarchical structure, resulting in a statistically elevated TAD hierarchy. Conversely, if the TAD hierarchy structure is not due to heterogeneous TAD superposition between the cells, then the mixing should result in a TAD hierarchy that is in between the two cells.

The results show that the second hypothesis in Figure B is more likely to be the mechanism of TAD hierarchy formation. This is a method to verify the TAD layer formation mechanism side by side, which is an indirect means under the influence of resolution and noise of current detection technology. Future real-time, high-resolution microimaging techniques may provide a more direct and intuitive demonstration.

Finally, the analogy relating to air conditioners is not intuitive or helpful. Numerous typos and lexical errors exist, for instance:

- Line 50: "conservative" → "conserved"

- Line 59: "Dynamic" → "Dynamics"
- Line 81: Perhaps, "as a possible mechanism"
- Line 117: A grammatical adjustment is necessary, potentially "features assume."
- Line 423: Adjustment may be "expression. Perhaps"

RESPONSE:

Thanks for your constructive suggestion. To make our analogy more intuitive, we have explained the colors of the graph, changed the filling of colors, and added legends such as arrows, enhancers, and boundary switches in the figure legend. The revised figure 7c, 7d are as follows:

Figure 7

Next, we would like to explain our innovation points and significance of the air-conditioning model. Currently, there are two main types of mechanism hypotheses for the formation of TAD hierarchies, including the loop-extrusion model, which focuses on the boundary formation mechanism and is unclear about the enhancer modulation mechanism within the ring, and the phase-separation model, which has difficulty in explaining enhancer influences across the boundary of TAD hierarchies. Our air-conditioning model, which highlights the diffuse enhancement of enhancers in hierarchical TAD, provides new insights into the mechanism of hierarchical TAD regulation.

Finally, we have checked and corrected all typos and lexical errors, and also increased the fluency of sentences in our manuscript.

Reviewer #3:

General:

In this paper, the authors performed a comprehensive benchmark study of the 13 algorithms to define TAD hierarchy. This work addressed an important research question. The results are solid and convincing. The authors provided some useful guidelines for practitioners how to select the optimal computational methods. The paper is also well written. I feel this work has potential to address broad interest from readers of Nature Communications, and researchers in 3D genome field. I just have a few comments and suggestions which may further improve the current manuscript. Here are my specific comments.

RESPONCE:

Thank you for acknowledging the value of our research and providing us with constructive suggestions. Based on your suggestions, we have made the following improvements to our manuscript.

Major comments:

1. The authors applied 13 methods to Hi-C data from 7 human cell lines (GSE63525), and only tested chr7. For the completeness of results, I hope the authors can perform genome-wide analysis to identify TAD hierarchy in all chromosomes. In addition to human data, I hope the authors can apply the 13 methods to mouse Hi-C data and drosophila Hi-C data, and evaluate whether their conclusions from the benchmark study can also be applied to mouse and drosophila.

Thank you for your constructive advice.

Indeed, we have collected Hi-C data from human cell lines and analyzed TAD hierarchy recognition algorithm in chromosome 7. It is important to evaluate the robustness of hierarchy recognition methods among different chromosomes and species. Therefore, we collected GM12878 (human) and CH12-LX (mouse) cell lines from GEO dataset with accession number GSE63525. Then, we downloaded S2 (drosophila) cell lines from GEO dataset with accession number GSE130778 and identified TAD hierarchy.

We first evaluated the similarity of various TAD hierarchy recognition methods on different chromosomes. We found that the identification effect characteristics of the method are similar across different chromosomes in mouse CH12-LX cells (Figure 5 for reviewer). For example, HiCKey and TADpole have low similarity with other methods, while Arrowhead and Armatous maintains high similarity. This suggests that the TAD hierarchy recognition effect is less affected by different chromosomes.

CH12-LX SSIM

Figure 5 for reviewer

Next, we applied the TAD hierarchy recognition methods to Hi-C data of mouse CH12-LX cell line and *Drosophila* S2 cell line. We noticed inconsistency of clustering pattern in GM12878, CH12-LX and S2 cell line (Figure 6 for reviewer). In different species, the SSIM of various TAD hierarchy identification methods is also greater than 0.8, but the clustering results are not completely consistent. This indicates that the algorithm has a certain degree of robustness. We speculated that the inconsistency of clustering results is due to differences in genome size and TAD size among different species.

Figure 6 for review

To verify the functional robustness of TAD boundaries identified by different methods, we compared the CTCF enrichment of TAD boundaries from different species. We noticed CTCF enrichment occurs at TAD boundaries in different species and we get the most enrichment in the OnTAD boundaries, (Figure 7 for review), proving the robustness of our conclusion.

Figure 7 for reviewer

To perform genome-wide analysis to identify TAD hierarchy in all chromosomes, we used 4 workstations and 5 server nodes to apply TAD hierarchy recognition methods (our increased computational resources are shown below). However, some algorithms, such as superTAD, consume too much time (>10 days per chromosome) and huge computational resources. It's hard to finish all 23 pairs of chromosome calculation in various species, resolutions, biological replicates, normalization methods and sequencing depths. Therefore, we apologized for not providing genome-wide benchmark. The robustness of recognition algorithms among chromosomes suggested the pattern we discovered on one chromosome may also be consistent across the entire genome.

Thanks again for your great suggestions, providing inspiration for us to verify the robustness of the results from a new perspective.

Computational resources:

A total of 4 workstations and 5 server nodes are used for data processing.

Workstation 1: The workstation used to test all methods had an Intel(R) Xeon(R) CPU W-2175 @ 2.50 GHz (19,712 KB cache size; 28 cores in total) and 256 GB of memory. The GPUs were one Nvidia Quadro RTX5000 16G. The operating system used was Ubuntu 20.04.2 LTS.

Workstation 2: The workstation used to test all methods had an Intel(R) Xeon(R) CPU W-2225 @ 4.10 GHz (8,448 KB cache size; 8 cores in total) and 256 GB of memory. The GPUs were one Radeon RX550/550X. The operating system used was Ubuntu 18.04.6 LTS.

Workstation 3: The workstation used to test all methods had an Intel(R) Xeon(R) Gold 5218R CPU @ 2.10 GHz (28,160 KB cache size; 40 cores in total) and 256 GB of memory. The GPUs were one NVIDIA RTX A6000 48 GB. The operating system used was Ubuntu 18.04.6 LTS.

Workstation 4: The workstation used to test all methods had an Intel(R) Xeon(R) Gold 6258R CPU @ 2.70 GHz (39,424 KB cache size; 112 cores in total) and 256 GB of memory. The GPUs were one Llvmpipe 2684. The operating system used was Ubuntu 18.04.6 LTS.

Server nodes 1: The server node used to test all methods had an Intel(R) Xeon(R) Gold 5320 CPU W-2175 @ 2.20 GHz (39,936 KB cache size; 52 cores in total) and 256 GB of memory. The GPUs were one Nvidia Tesla A100 40GB. The operating system used was Red Hat 4.8.5-44.

Server nodes 2, 3, 4, and 5 have the same specifications: The server node used to test all methods had an Intel(R) Xeon(R) Gold 5320 CPU W-2175 @ 2.20 GHz (39,936 KB cache size; 52 cores in total) and 256 GB of memory. The GPUs were two Nvidia Tesla A100 40GB. The operating system used was Red Hat 4.8.5-44.

2. Page 6, line 147, the authors claimed to benchmark 13 methods. However, Fig 2b, c, d only

contain 11 methods. TADtree and TADpole are missing. The authors need to add the results of TADtree TADpole in Fig 2 and other main figure / supplementary figures.

Thanks for your great suggestions, which is actually quite technically difficult, because these two methods consume a lot of resources and time to run the missing resolution data, as shown in the attached graph of their resource consumption, it takes about a week to run one chromosome at 10kb resolution with TADtree, which is extremely long. Moreover, in the past method development, most development teams gave a unified evaluation of the computational performance of the two methods, and the attached figure is also consistent with these evaluations. Therefore, it is technically difficult to complete this part of the evaluation and requires more time and computing resources.

3. In some figures (e.g. Fig 2c, Fig 7b, Fig S10, etc), I suggest the authors to present both Hi-C contact heatmap and inferred TAD hierarchy, to facilitate easy visual inspection and interpretation.

Thanks so much for your great advice!

It is true that arranging the Hi-C heat map with the TAD tier distribution map presents a more at-a-glance result, both in terms of the veracity of the data and the distribution of the tiers. This perfectly makes this part of the results clearer and easier to understand, and also somewhat avoids the algorithm's erroneous inference of the TAD strata. This is an excellent suggestion! We have modified the images of the corresponding results according to the reviewers' comments (Figure 2c, Figure 7b, Figure S10).

Figure 2c

Figure 7b

Figure S10

4. Fig 3, why some methods have less than 5 bin resolutions?

Thank you for careful review of our manuscript. Since not all methods work perfectly for all ranges of resolution, e.g., GMAP is difficult to produce results on 100kb data. Since 5kb and 10kb data are too fine and the input file size of individual samples is too large, the computation time of TADtree and TADpole,

which are already slow on low resolution, is extremely long. TADtree is already too long when predicting 25kb data. And for SuperTAD, it consumes too much computational resources, even from 50kb. We managed to run input files from part of chromosomes, while computing is almost inches away from 10kb. Therefore, when we do the comparison analysis, the results of GMAP(100kb), TADpole(5kb, 10kb), TADtree(5kb, 10kb, and 25kb), and SuperTAD(5kb, 10kb) are eliminated. The lack of resolution actually partially reflects the performance of the algorithms, both in terms of coverage of the resolution as well as in terms of computation time and computational resource consumption.

5. Methods section: in the definition of Hier_SSIM, the authors used a sliding window of 4Mb. I hope the authors can justify the selection of 4Mb. Whether the conclusions are robust for different size of sliding windows? Some sensitivity analysis would be very informative.

RESPONSE:

We thank you for meticulous review and mentioning this critical issue.

At first, we used 4Mb as the sliding window due to the fact that the size of the TAD is mainly enriched below 2Mb, and we chose two times the size of the TAD so as to ensure maximum retention of the feature information of the TAD hierarchical structure. We do lack in the assessment of the impact of the sliding window size on the robustness of the results, so we tested the impact on the clustering results using sliding window sizes of 1Mb, 2Mb, 3Mb, 4Mb, 5Mb, 6Mb, 7Mb, 8Mb, 9Mb, and 10Mb.

In order to quantify the effect of increasing sliding window size on the stability of the clustering results, we define the Iteration loss score (Iteration loss):

$$I^t = \frac{\sum_{x=i}^n \sum_{y=i}^n [\text{SSIM}(\mathbf{A}_x^t, \mathbf{A}_y^t) - \text{SSIM}(\mathbf{A}_x^{t-1}, \mathbf{A}_y^{t-1})]^2}{n^2}$$

where I^t represents the iterative loss under t window size, $t \in \{2\text{Mb}, 3\text{Mb}, 4\text{Mb}, 5\text{Mb}, 6\text{Mb}, 7\text{Mb}, 8\text{Mb}, 9\text{Mb}, 10\text{Mb}\}$. $\text{SSIM}(\mathbf{A}_x^t, \mathbf{A}_y^t)$ represents the correlation score between two TAD recognition results under t window size.

We test the effect of sliding window size on clustering results in application scenarios with different algorithms at the same resolution and different resolutions for the same algorithm, respectively (Figure 9 for reviewer).

Same resolution, different algorithms

Same algorithm, different resolutions

Figure 9 for reviewer

The results show that the sliding window tends to stabilize at 7-8 Mb. Therefore, we used 8 Mb as the sliding window size in the subsequent analyses. We thank the reviewer for great suggestions, which helped us to select a more appropriate sliding window size.

6. In the analysis related to Fig S8, the authors simulated the cellular heterogeneity by mixing GM12878 and K562. A better way is to create pseudo bulk Hi-C data from scHi-C data from GM12878 and IMR90, etc (PMID: 32946435), with different mixture proportions.

RESPONCE:

Thank you for constructive comments.

We strongly agree with your views. Using single-cell pseudo-bulk Hi-C data is a better way to verify our conclusion. Firstly, we downloaded and processed single cell Hi-C data from 4DN project (PMID: 32946435). Then we generated pseudo-bulk Hi-C data and compared with the bulk Hi-C data (Figure 10 for reviewer).

Figure 10 for reviewer

We observed relatively sparse data for single-cell Hi-C pseudo-bulk. But retained most of the features of bulk Hi-C, proving our data corrected.

Next, we mixed the single-cell Hi-C data of GM12878 and IMR90. To find the distribution of TAD hierarchical structure under different mixing ratios. Specifically, we generated pseudo-bulk Hi-C by taking the matrix of all single-cell Hi-C in the ratio of GM12878 cell number:IMR90 cell number = 0:10, 1:9, 2:8, 3:7, 4:6, 5:5, 6:4, 7:3, 8:2, 9:1, and 10:0. We then counted the TAD layer structure distributions under the 10kb, 25kb, 50kb, 100kb resolution, and the distribution of TAD hierarchical structure and number of boundaries.

Consistent with the experimental results of bulk Hi-C, the TAD hierarchy did not increase due to the mixing of cell lines, but decreased as the ratio of GM12878 decreased (Figure 11 for reviewer). It suggests that in the cellular heterogeneity system with the mixture of GM12878 and IMR90, the TAD hierarchical structure does not arise due to the superposition of heterogeneous cells, but rather, the TAD hierarchy is present in single cells.

Thank you for great suggestions, which provided us with a better dataset to validate our results.

Figure 11 for reviewer

7. What is the reproducibility of TAD hierarchy calling results? If applying each of 13 methods to biological replicates of Hi-C data from the same cell type, which methods can achieve the most reproducible results?

RESPONSE:

Thanks for your constructive advice.

It is an excellent ideal to measure the reproducibility of TAD hierarchy calling results on biological replicates of Hi-C data. We applied these methods to GM12878 Hi-C data, and found that matryoshka, SpectralTAD, OnTAD, Arrowhead, GMAP and HiCKey can achieve great reproducible results (Figure 12 for reviewer). This suggested that these methods are robust in identifying TAD in biological replicates.

And SpectralTAD can achieve the most reproducible results with 0.976539 similarity.

Figure 12 for reviewer

8. Fig 7b, since the authors compare normal cell line GM12878 with cancer cell line K562, whether the difference is due to structural variations, such as insertion, deletion, copy number variations, translocation, in K562 cells?

RESPONSE:

Thanks for your great question.

Indeed, structural variations, such as insertion, deletion, copy number variations, translocation, can cause abnormal Hi-C heatmap. And TAD hierarchy recognition algorithms are based on Hi-C heatmap. Thus, SVs can lead to TAD hierarchy recognition difference. Besides, abnormal transcription factors

concentration can lead to TAD boundary change. For example, CTCF loss can activate MYC gene expression.

To investigate whether the TAD hierarchy difference in Figure 7 is due to structural variations, we collected WGS (Whole Genome Sequencing) data of K562 cell line from ENCODE dataset with accession number ENCSR025GPQ (Figure 13 for reviewer). We noticed a copy number variation in the region, while no abnormal change in Hi-C map and TAD hierarchy. Then we zoomed in changed TAD hierarchy boundaries, and also did not find structural variations. This suggested that the TAD hierarchy difference between normal and cancer cell lines may not be caused by structural variations.

Figure 13 for reviewer

In conclusion, part of TAD hierarchy recognition difference is due to structural variations.

Minor comments:

1. The authors need to provide more details of running time and memory cost for each of 13 methods.

RESPONSE:

Thanks for your advice. We have provided the running time and memory cost as follow table (Supplementary Table 4):

		100kb	50kb	25kb	10kb	5kb
Arrowhead	time	0m14.489s	0m24.141s	0m35.258s	1m6.845s	2m2.393s
	memory	2.0G	3.9G	6.7G	9.9G	10.9G
Armatus	time	0m7.490s	0m22.973s	0m53.060s	6m48.109s	32m58.254s
	memory	29.6M	116.9M	465.7M	2.8G	11.3G
TADtree	time	6m44.323s	62m27.921s	2274m51.231s		
	memory	136.7M	371.1M	499.2M		
GMAP	time		~20s	~1m	~3m	~10m
	memory		1.2G	2.2G	18.0G	27.9G
HiTAD	time	10.579s	22.462s	53.770s	5m10.099s	17m39.057s
	memory	598.8M	1.3G	1.6G	3.2G	4.4G
deDoc	time	0m22.88s	1m17.34s	4m18.36s	18m22.21s	66m0.42s
	memory	6.5G	12.3G	19.6G	26.7G	31.9G
matryoshka	time	0m7.117s	0m32.810s	2m35.674s	25m6.582s	80m33.075s
	memory	39.5M	118.3M	465.8M	2.8G	11.3G
OnTAD	time	0m0.502s	0m1.342s	0m4.492s	0m27.758s	1m59.385s
	memory	53.9M	235.1M	1.8G	11.4G	45.5G
TADpole	time	46m53.274s	113m26.482s	257m9.550s		
	memory	927.7M	2.8G	5.1G		
SpectralTAD	time	~5s	~10s	~15s	~30s	~1m
	memory	984.8M	1.2G	2.0G	6.10G	24.2G
HiCKey	time	0m0.653s	0m1.882s	0m4.289s	0m8.063s	0m11.179s
	memory	39.5M	99.0M	139.2M	275.4M	433.8M
SuperTAD	time	917m48.012s	8592m16.054s			
	memory	11.2G	88.4G			
GRiNCH	time	0m23s	0m48s	4m23s	10m22s	47m57s
	memory	522M	1750M	6326M	37154M	145508M

Supplementary Table 4

The cells with slash main the computational limitation in resolution coverage of method (GMAP could not detect matrix with resolution lower than 100 Kb) or overloaded computing memory(TADtree, TADpole, and SuperTAD).

2. It seems that Fig 7c is missing?

RESPONSE:

Thanks for reminding us. We speculated that Fig 7c is missing because of image format. Therefore, we changed image format. And if the Fig7c is still missing in new submitted manuscript, please do not hesitate to communicate with the editors and us, and we will try to submit the manuscript to you in a timely

manner by other means. We have taken a screenshot of Figure 7c as follows:

Figure 7c

3. Typo. Page 8, line 210: "with 2M" should be "within 2Mb". Line 211: "within 1M" should be "within 1Mb".

RESPONSE:

Thanks for your suggestion. We have corrected the errors according to your suggestions and check the entire manuscript.

4. Typo. Legend in Fig 5d "Raio" should be "Ratio".

RESPONSE:

Thanks for your detailed review and suggestion. We have corrected the errors according to your suggestions and check the entire manuscript.

5. The authors considered 5 bin resolutions (5Kb, 10Kb, 25Kb, 50Kb, 100Kb). Page 11, line 290: matryoshka also performs well on low-resolution data (~500Kb). Is "500Kb" a typo of "50Kb"?

RESPONSE:

Thanks for your question. Actually, the "500Kb" is written correctly. In Hi-C heatmap, resolution is defined as one pixel representing chromatin length, so the "500Kb" is lower than "50Kb" in resolution.

Reviewer #1 (Remarks to the Author):

The authors have addressed all my comments.

Reviewer #1 (Remarks on code availability):

Since it is a review work, the sources for all the tools used have been provided in the repository.

Reviewer #2 (Remarks to the Author):

The air conditioner model is still not clear enough to the readers. From the text it seems to suggest that the TADs / TAD boundaries are the air conditioners since "the high-level TAD boundary with active regulatory factors plays a similar role as a central air conditioner in a large room or corridor". But Fig 7d now seems to suggest that the boundaries are not air conditioners but the "doors"?

Additionally, the text reads that 1) 'the presence and absence of gene-specific enhancers can be analogized to two states: "on" and "off" '; and that 2) 'a gene-specific enhancer is like a fan'. This creates a confusion about whether the presence/absence of the enhancer is the on/off as suggested by 1) or the regulation of the present enhancer controls the on/off as suggested by 2) and fig 7d. It would be very helpful to make that clear.

Finally, I appreciate that the authors used the matching color (dark green) for the genes in Fig 7c and 7d. But it's very difficult to intuitively notice the analogy between 7c and 7d.

Reviewer #3 (Remarks to the Author):

The authors did a great job in the revision and fully addressed my previous comments and suggestions. The revised manuscript has been significantly improved. I don't have additional comments and feel the current manuscript is ready for acceptance.

REVIEWERS' COMMENTS

Reviewer #1 (Remarks to the Author):

The authors have addressed all my comments.

RESPONCE:

Thank you for your careful review of our manuscript and for your very pertinent and valuable suggestions, the quality of the article has been greatly improved after revision in accordance with your constructive suggestions.

Reviewer #1 (Remarks on code availability):

Since it is a review work, the sources for all the tools used have been provided in the repository.

RESPONCE:

Sure, we've uploaded the source and relevant code for all the tools included in the review to Github and linked to Zenodo. You can access them via the links below:

1. Github: https://github.com/XiangXuCode/TAD_hierarchy_benchmark
2. Zenodo: <https://zenodo.org/records/10982207>

Reviewer #2 (Remarks to the Author):

The air conditioner model is still not clear enough to the readers. From the text it seems to suggest that the TADs / TAD boundaries are the air conditioners since "the high-level TAD boundary with active regulatory factors plays a similar role as a central air conditioner in a large room or corridor". But Fig 7d now seems to suggest that the boundaries are not air conditioners but the "doors"?

Additionally, the text reads that 1) 'the presence and absence of gene-specific enhancers can be analogized to two states: "on" and "off" '; and that 2) 'a gene-specific enhancer is like a fan'. This creates a confusion about whether the presence/absence of the enhancer is the on/off as suggested by 1) or the regulation of the present enhancer controls the on/off as suggested by 2) and fig 7d. It would be very helpful to make that clear.

Finally, I appreciate that the authors used the matching color (dark green) for the genes in Fig 7c and 7d. But it's very difficult to intuitively notice the analogy between 7c and 7d.

RESPONCE:

Thank you very much for your valuable and pertinent comments on our manuscript, and I apologize for the confusion caused by not clearly articulating the air conditioner model in this part of the original paper. To show our model more intuitively, we revised the Fig 7c and 7d and explain it in detail here:

In our model, the air conditioners represent enhancers. The yellow DNA fragment represents high-

level TAD boundary, which mediates the formation of three TADs (Figure 1 for reviewer left). In our research, active modifications and gene transcription were enriched at the TAD boundary. Therefore, we marked the genes represented by gene circles and the enhancers represented by red air conditioners on the TAD boundaries (Figure 1 for reviewer right).

As shown in Figure 1 for reviewer, high-level TAD boundary gathered a number of enhancers. Since enhancers play the role in activating transcription by near space interaction, there is an analogy between enhancer and air conditioner. The more concentrated the distribution of enhancers, the stronger activation of genes.

In different cells types, states or conditions, TAD hierarchy might change. The reduction of TAD boundary level leads to reduction of enhancer concentration (Figure 2 for reviewer). The reduction of enhancer concentration limits the activation of gene expression. This is consistent with our model that the concentration of air conditioner affects the ability of controlling the temperature of aggregation area.

Thanks again for your excellent advice. We have revised the figures and manuscript to better explain our model.

Figure 1 for reviewer

Figure 2 for reviewer

Reviewer #3 (Remarks to the Author):

The authors did a great job in the revision and fully addressed my previous comments and suggestions. The revised manuscript has been significantly improved. I don't have additional comments and feel the current manuscript is ready for acceptance.

RESPONSE:

Thank you for reviewing our manuscript and for your constructive comments. We have revised the manuscript according to your suggestions, and the overall quality and level of the article has been greatly improved and is more in line with the standard of the journal.